# TILTED EMPIRICAL RISK MINIMIZATION

**Tian Li**[*]
CMU
tianli@cmu.edu

**Ahmad Beirami**[*]
Facebook AI
beirami@fb.com

**Maziar Sanjabi**
Facebook AI
maziars@fb.com

**Virginia Smith**
CMU
smithv@cmu.edu

## ABSTRACT

Empirical risk minimization (ERM) is typically designed to perform well on the average loss, which can result in estimators that are sensitive to outliers, generalize poorly, or treat subgroups unfairly. While many methods aim to address these problems individually, in this work, we explore them through a unified framework—tilted empirical risk minimization (TERM). In particular, we show that it is possible to flexibly tune the impact of individual losses through a straightforward extension to ERM using a hyperparameter called the tilt. We provide several interpretations of the resulting framework: We show that TERM can increase or decrease the influence of outliers, respectively, to enable fairness or robustness; has variance-reduction properties that can benefit generalization; and can be viewed as a smooth approximation to a superquantile method. We develop batch and stochastic first-order optimization methods for solving TERM, and show that the problem can be efficiently solved relative to common alternatives. Finally, we demonstrate that TERM can be used for a multitude of applications, such as enforcing fairness between subgroups, mitigating the effect of outliers, and handling class imbalance. TERM is not only competitive with existing solutions tailored to these individual problems, but can also enable entirely new applications, such as simultaneously addressing outliers and promoting fairness.

## 1 INTRODUCTION

Many statistical estimation procedures rely on the concept of empirical risk minimization (ERM), in which the parameter of interest, $\theta \in \Theta \subseteq \mathbb{R}^d$, is estimated by minimizing an average loss over the data:

$$\overline{R}(\theta) := \frac{1}{N} \sum_{i \in [N]} f(x_i; \theta).$$ 

(1)

While ERM is widely used and has nice statistical properties, it can perform poorly in situations where average performance is not an appropriate surrogate for the problem of interest. Significant research has thus been devoted to developing alternatives to traditional ERM for diverse applications, such as learning in the presence of noisy/corrupted data (Jiang et al., 2018; Khetan et al., 2018), performing classification with imbalanced data (Lin et al., 2017; Malisiewicz et al., 2011), ensuring that subgroups within a population are treated fairly (Hashimoto et al., 2018; Samadi et al., 2018), or developing solutions with favorable out-of-sample performance (Namkoong & Duchi, 2017).

In this paper, we suggest that deficiencies in ERM can be flexibly addressed via a unified framework, *tilted empirical risk minimization (TERM)*. TERM encompasses a family of objectives, parameterized by a real-valued hyperparameter, $t$. For $t \in \mathbb{R}^{\backslash 0}$, the $t$-tilted loss (TERM objective) is given by:

$$\widetilde{R}(t; \theta) := \frac{1}{t} \log \left( \frac{1}{N} \sum_{i \in [N]} e^{tf(x_i; \theta)} \right).$$ 

(2)

TERM generalizes ERM as the 0-tilted loss recovers the average loss, i.e., $\widetilde{R}(0, \theta) = \overline{R}(\theta)$.[1] It also recovers other popular alternatives such as the max-loss ($t \to +\infty$) and min-loss ($t \to -\infty$) (Lemma 2). For $t > 0$, the objective is a common form of exponential smoothing, used to approximate the max (Kort & Bertsekas, 1972; Pee & Royset, 2011). Variants of tilting have been studied in several contexts,

---

[*]Equal contribution.

[1]$\widetilde{R}(0; \theta)$ is defined in (14) via the continuous extension of $R(t; \theta)$.

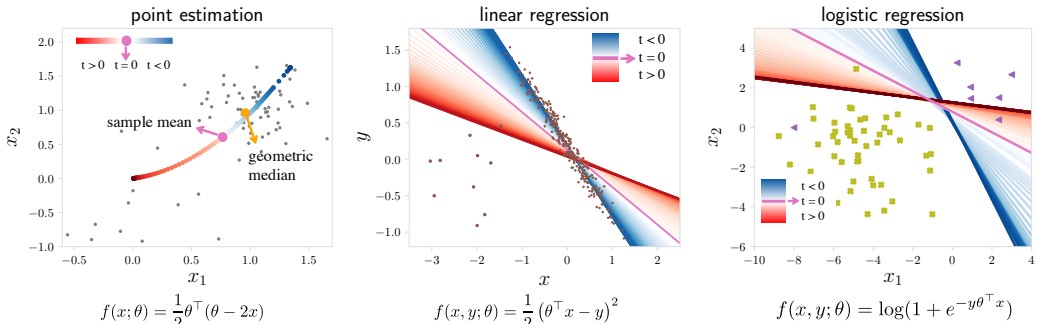

Figure 1: Toy examples illustrating TERM as a function of $t$: (a) finding a point estimate from a set of 2D samples, (b) linear regression with outliers, and (c) logistic regression with imbalanced classes. While positive values of $t$ magnify outliers, negative values suppress them. Setting $t=0$ recovers the original ERM objective (1).

including robust regression (Wang et al., 2013) ($t<0$), importance sampling (Wainwright et al., 2005), sequential decision making (Howard & Matheson, 1972; Nass et al., 2019), and large deviations theory (Beirami et al., 2018). However, despite the rich history of tilted objectives, they have not seen widespread use in machine learning. In this work, we aim to bridge this gap by: (i) rigorously studying the objective in a general form, and (ii) exploring its utility for a number of ML applications. Surprisingly, we find that this simple extension to ERM is competitive for a wide range of problems.

To highlight how the TERM objective can help with issues such as outliers or imbalanced classes, we discuss three motivating examples below, which are illustrated in Figure 1.

*(a) Point estimation:* As a first example, consider determining a point estimate from a set of samples that contain some outliers. We plot an example 2D dataset in Figure 1a, with data centered at (1,1). Using traditional ERM (i.e., TERM with $t = 0$) recovers the *sample mean*, which can be biased towards outlier data. By setting $t < 0$, TERM can suppress outliers by reducing the relative impact of the largest losses (i.e., points that are far from the estimate) in (2). A specific value of $t < 0$ can in fact approximately recover the geometric median, as the objective in (2) can be viewed as approximately optimizing specific loss quantiles (a connection which we make explicit in Section 2). In contrast, if these 'outlier' points are important to estimate, setting $t > 0$ will push the solution towards a point that aims to minimize variance, as we prove more rigorously in Section 2, Theorem 4.

*(b) Linear regression:* A similar interpretation holds for the case of linear regression (Figure 2b). As $t \to -\infty$, TERM finds a line of best while ignoring outliers. However, this solution may not be preferred if we have reason to believe that these 'outliers' should not be ignored. As $t \to +\infty$, TERM recovers the min-max solution, which aims to minimize the worst loss, thus ensuring the model is a reasonable fit for *all* samples (at the expense of possibly being a worse fit for many). Similar criteria have been used, e.g., in defining notions of fairness (Hashimoto et al., 2018; Samadi et al., 2018). We explore several use-cases involving robust regression and fairness in more detail in Section 5.

*(c) Logistic regression:* Finally, we consider a binary classification problem using logistic regression (Figure 2c). For $t \in \mathbb{R}$, the TERM solution varies from the nearest cluster center ($t \to -\infty$), to the logistic regression classifier ($t=0$), towards a classifier that magnifies the misclassified data ($t \to +\infty$). We note that it is common to modify logistic regression classifiers by adjusting the decision threshold from 0.5, which is equivalent to moving the intercept of the decision boundary. This is fundamentally different than what is offered by TERM (where the slope is changing). As we show in Section 5, this added flexibility affords TERM with competitive performance on a number of classification problems, such as those involving noisy data, class imbalance, or a combination of the two.

**Contributions.** In this work, we explore TERM as a simple, unified framework to flexibly address various challenges with empirical risk minimization. We first analyze the objective and its solutions, showcasing the behavior of TERM with varying $t$ (Section 2). Our analysis provides novel connections between tilted objectives and superquantile methods. We develop efficient methods for solving TERM (Section 4), and show via numerous case studies that TERM is competitive with existing, problem-specific state-of-the-art solutions (Section 5). We also extend TERM to handle compound issues, such as the simultaneous existence of noisy samples and imbalanced classes (Section 3). Our results demonstrate the effectiveness and versatility of tilted objectives in machine learning.

## 2    TERM: PROPERTIES & INTERPRETATIONS

To better understand the performance of the $t$-tilted losses in (2), we provide several interpretations of the TERM solutions, leaving the full statements of theorems and proofs to the appendix. We make no distributional assumptions on the data, and study properties of TERM under the assumption that the loss function forms a generalized linear model, e.g., $L_2$ loss and logistic loss (Appendix D). However, we also obtain favorable empirical results using TERM with other objectives such as deep neural networks and PCA in Section 5, motivating the extension of our theory beyond GLMs in future work.

**General properties.** We begin by noting several general properties of the TERM objective (2). Given a smooth $f(x; \theta)$, the $t$-tilted loss is smooth for all finite $t$ (Lemma 4). If $f(x; \theta)$ is strongly convex, the $t$-tilted loss is strongly convex for $t > 0$ (Lemma 5). We visualize the solutions to TERM for a toy problem in Figure 2, which allows us to illustrate several special cases of the general framework. As discussed in Section 1, TERM can recover traditional ERM ($t$=0), the max-loss ($t\to+\infty$), and the min-loss ($t\to-\infty$). As we demonstrate in Section 5, providing a smooth tradeoff between these specific losses can be beneficial for a number of practical use-cases—both in terms of the resulting solution and the difficulty of solving the problem itself. Interestingly, we additionally show that the TERM solution can be viewed as a smooth approximation to a *superquantile* method, which aims to minimize quantiles of losses such as the median loss. In Figure 2, it is clear to see why this may be beneficial, as the median loss (orange) can be highly non-smooth in practice. We make these rough connections more explicit via the interpretations below.

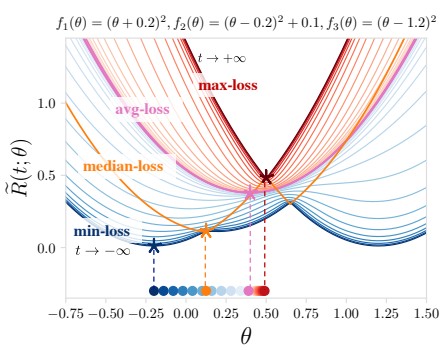

Figure 2: TERM objectives for a squared loss problem with $N = 3$. As $t$ moves from $-\infty$ to $+\infty$, $t$-tilted losses recover min-loss, avg-loss, and max-loss. TERM is smooth for all finite $t$ and convex for positive $t$.

**(Interpretation 1) Re-weighting samples to magnify/suppress outliers.** As discussed via the toy examples in Section 1, the TERM objective can be tuned (using $t$) to magnify or suppress the influence of outliers. We make this notion rigorous by exploring the *gradient* of the $t$-tilted loss in order to reason about the solutions to the objective defined in (2).

**Lemma 1** (Tilted gradient, proof in Appendix B). *For a smooth loss function $f(x; \theta)$,*

$$\nabla_\theta \widetilde{R}(t;\theta) = \sum_{i \in [N]} w_i(t;\theta) \nabla_\theta f(x_i;\theta), \; where \; w_i(t;\theta) := \frac{e^{tf(x_i;\theta)}}{\sum_{j \in [N]} e^{tf(x_j;\theta)}} = \frac{1}{N} e^{t(f(x_i;\theta) - \widetilde{R}(t;\theta))}. \quad (3)$$

From this, we can observe that the tilted gradient is a weighted average of the gradients of the original individual losses, where each data point is weighted exponentially proportional to the value of its loss. Note that $t = 0$ recovers the uniform weighting associated with ERM, i.e., $w_i(t; \theta) = 1/N$. For positive $t$, it *magnifies* the outliers—samples with large losses—by assigning more weight to them, and for negative $t$, it *suppresses* the outliers by assigning less weight to them.

**(Interpretation 2) Tradeoff between average-loss and min/max-loss.** To put *Interpretation 1* in context and understand the limits of TERM, a benefit of the framework is that it offers a continuum of solutions between the min and max losses. Indeed, for positive values of $t$, TERM enables a smooth tradeoff between the average-loss and max-loss (as we demonstrate in Figure 10, Appendix I). Hence, TERM can selectively improve the worst-performing losses by paying a penalty on average performance, thus promoting a notion of uniformity or fairness (Hashimoto et al., 2018). On the other hand, for negative $t$, the solutions achieve a smooth tradeoff between average-loss and min-loss, which can have the benefit of focusing on the 'best' losses, or ignoring outliers (Theorem 3, Appendix D).

**Theorem** (Formal statement and proof in Appendix D, Theorem 3). *Let $\breve{\theta}(t)$ be the minimizer of $\widetilde{R}(t;\theta)$, referred to as $t$-tilted solution. Then, for $t > 0$, max-loss, $\widehat{R}(\breve{\theta}(t))$, is non-increasing with $t$ while the average loss, $\overline{R}(\breve{\theta}(t))$, is non-decreasing with $t$.*

**(Interpretation 3) Empirical bias/variance tradeoff.** Another key property of the TERM solutions is that the *empirical variance* of the loss across all samples decreases as $t$ increases (Theorem 4).

Hence, by increasing $t$, it is possible to trade off between optimizing the average loss vs. reducing variance, allowing the solutions to potentially achieve a better bias-variance tradeoff for generalization (Bennett, 1962; Hoeffding, 1994; Maurer & Pontil, 2009) (Figure 10, Appendix I). We use this property to achieve better generalization in classification in Section 5. We also prove that the cosine similarity between the loss vector and the all-ones vector monotonically increases with $t$ (Theorem 5), which shows that larger $t$ promotes a more *uniform* performance across all losses and can have implications for fairness defined as representation disparity (Hashimoto et al., 2018) (Section 5.2).

**Theorem** (Formal statement and proof in Appendix D, Theorem 4). *Let* $\mathbf{f}(\theta) := (f(x_1; \theta)), \ldots, f(x_N; \theta))$ *be the loss vector for parameter* $\theta$. *Then, the variance of the vector* $\mathbf{f}(\breve{\theta}(t))$ *is non-increasing with* $t$ *while its average, i.e.,* $\overline{R}(\breve{\theta}(t))$, *is non-decreasing with* $t$.

**(Interpretation 4) Approximate superquantile method.** Finally, we show that TERM is related to *superquantile*-based objectives, which aim to minimize specific quantiles of the individual losses that exceed a certain value (Rockafellar et al., 2000). For example, optimizing for 90% of the individual losses (ignoring the worst-performing 10%) could be a more reasonable practical objective than the pessimistic min-max objective. Another common application of this is to use the median in contrast to the mean in the presence of noisy outliers. As we discuss in Appendix G, superquantile methods can be reinterpreted as minimizing the $k$-loss, defined as the $k$-th smallest loss of $N$ (i.e., 1-loss is the min-loss, $N$-loss is the max-loss, $(N-1)/2$-loss is the median-loss). While minimizing the $k$-loss is more desirable than ERM in many applications, the $k$-loss is non-smooth (and generally non-convex), and is challenging to solve for large-scale problems (Jin et al., 2020; Nouiehed et al., 2019b).

**Theorem** (Formal statement and proof in Appendix G, Theorem 10). *The quantile of the losses that exceed a given value is upper bounded by a smooth function of the TERM objective. Further, the $t$-tilted solutions are good approximate solutions of the superquantile ($k$-loss) optimization.*

## 3 TERM EXTENDED: HIERARCHICAL MULTI-OBJECTIVE TILTING

We also consider an extension of TERM that can be used to address practical applications requiring multiple objectives, e.g., simultaneously achieving robustness to noisy data *and* ensuring fair performance across subgroups. Existing approaches typically aim to address such problems in isolation. To handle multiple objectives with TERM, let each sample $x$ be associated with a group $g \in [G]$, i.e., $x \in g$. These groups could be related to the labels (e.g., classes in a classification task), or may depend only on features. For any $t, \tau \in \mathbb{R}$, we define multi-objective TERM as:

$$\widetilde{J}(t, \tau; \theta) := \frac{1}{t} \log \left( \frac{1}{N} \sum_{g \in [G]} |g| e^{t \widetilde{R}_g(\tau; \theta)} \right), \quad \text{where} \quad \widetilde{R}_g(\tau; \theta) := \frac{1}{\tau} \log \left( \frac{1}{|g|} \sum_{x \in g} e^{\tau f(x; \theta)} \right), \quad (4)$$

and $|g|$ is the size of group $g$. Multi-objective TERM recovers sample-level TERM as a special case for $\tau = t$ (Appendix, Lemma 7), and reduces to group-level TERM with $\tau \to 0$. Note that all properties discussed in Section 2 carry over to group-level TERM. Similar to the tilted gradient (3), the multi-objective tilted gradient is a weighted sum of the gradients (Appendix, Lemma 6), making it similarly efficient to solve. We validate the effectiveness of hierarchical tilting empirically in Section 5.3, where we show that TERM can significantly outperform baselines to handle class imbalance *and* noisy outliers simultaneously.

## 4 SOLVING TERM

To solve TERM, we suggest batch and stochastic variants of traditional first-order gradient-based optimization methods. TERM in the batch setting (Batch TERM) is summarized in Algorithm 1 in the context of solving multi-objective hierarchical TERM (4) for full generality. The main steps include computing the tilted gradients of the hierarchical objective defined in (4). Note that Batch TERM with $t = \tau$ reduces to solving the sample-level tilted objective (2). We also provide a stochastic variant in Algorithm 2, Appendix H. At a high level, at each iteration, group-level tilting is addressed by choosing a group based on the tilted weight vector estimated via stochastic dynamics. Sample-level tilting is then incorporated by re-weighting the samples in a uniformly drawn mini-batch. We find that these methods perform well empirically on a variety of tasks (Section 5).

---

**Algorithm 1:** Batch TERM

---

**Input:** $t, \tau, \alpha$
**while** *stopping criteria not reached* **do**
    **for** $g \in [G]$ **do**
        compute the loss $f(x; \theta)$ and gradient $\nabla_\theta f(x; \theta)$ for all $x \in g$
        $\widetilde{R}_{g,\tau} \leftarrow \tau$-tilted loss (4) on group $g$, $\nabla_\theta \widetilde{R}_{g,\tau} \leftarrow \frac{1}{|g|} \sum_{x \in g} e^{\tau f(x;\theta) - \tau \widetilde{R}_{g,\tau}} \nabla_\theta f(x; \theta)$
    **end**
    $\widetilde{J}_{t,\tau} \leftarrow \frac{1}{t} \log \left( \frac{1}{N} \sum_{g \in [G]} |g| e^{t \widetilde{R}_g(\tau;\theta)} \right)$, $w_{t,\tau,g} \leftarrow |g| e^{t \widetilde{R}_{\tau,g} - t \widetilde{J}_{t,\tau}}$
    $\theta \leftarrow \theta - \frac{\alpha}{N} \sum_{g \in [G]} w_{t,\tau,g} \nabla_\theta \widetilde{R}_{g,\tau}$
**end**

---

We defer readers to Appendix H for general properties of TERM (smoothness, convexity) that may vary with $t$ and affect the convergence of gradient-based methods used to solve the objective.

## 5 TERM IN PRACTICE: USE CASES

In this section, we showcase the flexibility, wide applicability, and competitive performance of the TERM framework through empirical results on a variety of real-world problems such as handling outliers (Section 5.1), ensuring fairness and improving generalization (Section 5.2), and addressing compound issues (Section 5.3). Despite the relatively straightforward modification TERM makes to traditional ERM, we show that $t$-tilted losses not only outperform ERM, but either outperform or are competitive with state-of-the-art, problem-specific tailored baselines on a wide range of applications.

We provide implementation details in Appendix J. All code, datasets, and experiments are publicly available at `github.com/litian96/TERM`. For experiments with positive $t$ (Section 5.2), we tune $t \in \{0.1, 0.5, 1, 5, 10, 50, 100, 200\}$ on the validation set. In our initial robust regression experiments, we find that the performance is robust to various $t$'s, and we thus use a fixed $t = -2$ for all experiments involving negative $t$ (Section 5.1 and Section 5.3). For all values of $t$ tested, the number of iterations required to solve TERM is within $2\times$ that of standard ERM.

### 5.1 MITIGATING NOISY OUTLIERS

We begin by investigating TERM's ability to find robust solutions that reduce the effect of noisy outliers. We note that we specifically focus on the setting of 'robustness' involving random additive noise; the applicability of TERM to more adversarial forms of robustness would be an interesting direction of future work. We do not compare with approaches that require additional clean validation data (e.g., Hendrycks et al., 2018; Ren et al., 2018; Roh et al., 2020; Veit et al., 2017), as such data can be costly to obtain in practice.

**Robust regression.** We first consider a regression task with noise corrupted targets, where we aim to minimize the root mean square error (RMSE) on samples from the Drug Discovery dataset (Diakoniko-las et al., 2019; Olier et al., 2018). The task is to predict the bioactivities given a set of chemical compounds. We compare against linear regression with an $L_2$ loss, which we view as the 'standard' ERM solution for regression, as well as with losses commonly used to mitigate outliers—the $L_1$ loss and Huber loss (Huber, 1964). We also compare with consistent robust regression (CRR) (Bhatia et al., 2017) and STIR (Mukhoty et al., 2019), recent state-of-the-art methods specifically designed for label noise in robust regression. In this particular problem, TERM is equivalent to exponential squared loss, studied in (Wang et al., 2013). We apply TERM at the sample level with an $L_2$ loss, and generate noisy outliers by assigning random targets drawn from $\mathcal{N}(5, 5)$ on a fraction of the samples.

In Table 1, we report RMSE on clean test data for each objective and under different noise levels. We also present the performance of an oracle method (Genie ERM) which has access to all of the clean data samples with the noisy samples removed. *Note that Genie ERM is not a practical algorithm and is solely presented to set the expected performance limit in the noisy setting.* The results indicate that TERM is competitive with baselines on the 20% noise level, and achieves better robustness with moderate-to-extreme noise. We observe similar trends in scenarios involving both

noisy features and targets (Appendix I.2). CRR tends to run slowly as it scales cubicly with the number of dimensions (Bhatia et al., 2017), while solving TERM is roughly as efficient as ERM.

Table 1: TERM is competitive with robust *regression* baselines, particularly in high noise regimes.

| objectives | test RMSE (Drug Discovery) | | |
|---|---|---|---|
| | 20% noise | 40% noise | 80% noise |
| ERM | 1.87 (.05) | 2.83 (.06) | 4.74 (.06) |
| $L_1$ | **1.15** (.07) | 1.70 (.12) | 4.78 (.08) |
| Huber (Huber, 1964) | **1.16** (.07) | 1.78 (.11) | 4.74 (.07) |
| STIR (Mukhoty et al., 2019) | **1.16** (.07) | 1.75 (.12) | 4.74 (.06) |
| CRR (Bhatia et al., 2017) | **1.10** (.07) | 1.51 (.08) | 4.07 (.06) |
| TERM | **1.08** (.05) | **1.10** (.04) | **1.68** (.03) |
| Genie ERM | 1.02 (.04) | 1.07 (.04) | 1.04 (.03) |

Table 2: TERM is competitive with robust *classification* baselines, and is superior in high noise regimes.

| objectives | test accuracy (CIFAR10, Inception) | | |
|---|---|---|---|
| | 20% noise | 40% noise | 80% noise |
| ERM | 0.775 (.004) | 0.719 (.004) | 0.284 (.004) |
| RandomRect (Ren et al., 2018) | 0.744 (.004) | 0.699 (.005) | 0.384 (.005) |
| SelfPaced (Kumar et al., 2010) | 0.784 (.004) | 0.733 (.004) | 0.272 (.004) |
| MentorNet-PD (Jiang et al., 2018) | 0.798 (.004) | 0.731 (.004) | 0.312 (.005) |
| GCE (Zhang & Sabuncu, 2018) | **0.805** (.004) | 0.750 (.004) | 0.433 (.005) |
| TERM | 0.795 (.004) | **0.768** (.004) | **0.455** (.005) |
| Genie ERM | 0.828 (.004) | 0.820 (.004) | 0.792 (.004) |

Note that the outliers considered here are unstructured with random noise, and not adversarial. This makes it possible for the methods to find the underlying structure of clean data even if the majority of the samples are noisy outliers. To gain more intuition on these cases, we also generate synthetic two-dimensional data points and test the performance of TERM under 0%, 20%, 40%, and 80% noise for linear regression. TERM with $t = -2$ performs well in all noise levels (Figure 11 and 12 in Appendix I.2). However, as might be expected, in Figure 14 (Appendix I.2) we show that TERM may overfit to noisy samples when the noise is structured and the noise values are large (e.g., 80%).

**Robust classification.** It is well-known that deep neural networks can easily overfit to corrupted labels (e.g., Zhang et al., 2017). While the theoretical properties we study for TERM (Section 2) do not directly cover objectives with neural network function approximations, we show that TERM can be applied empirically to DNNs to achieve robustness to noisy training labels. MentorNet (Jiang et al., 2018) is a popular method in this setting, which learns to assign weights to samples based on feedback from a student net. Following the setup in Jiang et al. (2018), we explore classification on CIFAR10 (Krizhevsky et al., 2009) when a fraction of the training labels are corrupted with uniform noise—comparing TERM with ERM and several state-of-the-art approaches (Krizhevsky et al., 2009; Kumar et al., 2010; Ren et al., 2018; Zhang & Sabuncu, 2018). As shown in Table 2, TERM performs competitively with 20% noise, and outperforms all baselines in the high noise regimes. We use MentorNet-PD as a baseline since it does not require clean validation data. In Appendix I.2, we show that TERM also matches the performance of MentorNet-DD, which requires clean validation data. To help reason about the performance of TERM, we also explore a simpler, two-dimensional logistic regression problem in Figure 13, Appendix I.2, finding that TERM with $t=-2$ is similarly robust across the considered noise regimes.

**Low-quality annotators.** It is not uncommon for practitioners to obtain human-labeled data for their learning tasks from crowd-sourcing platforms. However, these labels are usually noisy in part due to the varying quality of the human annotators. Given a collection of labeled samples from crowd-workers, we aim to learn statistical models that are robust to the potentially low-quality annotators. As a case study, following the setup of (Khetan et al., 2018), we take the CIFAR-10 dataset and simulate 100 annotators where 20 of them are *hammers* (i.e., always correct) and 80 of them are *spammers* (i.e., assigning labels uniformly at random). We apply TERM at the annotator group level in (4), which is equivalent to assigning annotator-level weights based on the aggregate value of their loss. As shown in Figure 3, TERM is able to achieve the test accuracy limit set by *Genie ERM*, i.e., *the ideal performance obtained by completely removing the known outliers*. We note in particular that the accuracy reported by (Khetan et al., 2018) (0.777) is lower than TERM (0.825) in the same setup, even though their approach is a two-pass algorithm requiring at least to double the training time. We provide full empirical details and investigate additional noisy annotator scenarios in Appendix I.2.

## 5.2 Fairness and Generalization

In this section, we show that positive values of $t$ in TERM can help promote fairness (e.g., via learning fair representations), and offer variance reduction for better generalization.

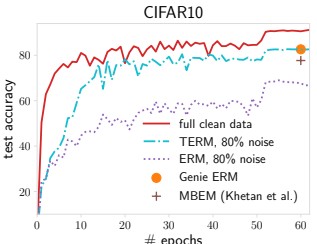
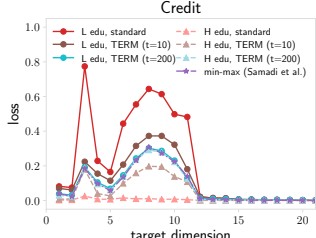
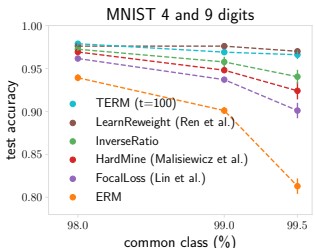

Figure 3: TERM ($t=-2$) completely removes the impact of noisy annotators, reaching the performance limit set by Genie ERM.

Figure 4: TERM-PCA flexibly trades the performance on the high (H) edu group for the performance on the low (L) edu group.

Figure 5: TERM ($t=100$) is competitive with state-of-the-art methods for classification with imbalanced classes.

**Fair principal component analysis (PCA).** We explore the flexibility of TERM in learning fair representations using PCA. In fair PCA, the goal is to learn low-dimensional representations which are fair to all considered subgroups (e.g., yielding similar reconstruction errors) (Kamani et al., 2019; Samadi et al., 2018; Tantipongpipat et al., 2019). Despite the non-convexity of the fair PCA problem, we apply TERM to this task, referring to the resulting objective as TERM-PCA. We tilt the same loss function as in (Samadi et al., 2018): $f(X; U) = \frac{1}{|X|} \left( \|X - XUU^\top\|_F^2 - \|X - \hat{X}\|_F^2 \right)$, where $X \in \mathbb{R}^{n \times d}$ is a subset (group) of data, $U \in \mathbb{R}^{d \times r}$ is the current projection, and $\hat{X} \in \mathbb{R}^{n \times d}$ is the optimal rank-$r$ approximation of $X$. Instead of solving a more complex min-max problem using semi-definite programming as in (Samadi et al., 2018), which scales poorly with problem dimension, we apply gradient-based methods, re-weighting the gradients at each iteration based on the loss on each group. In Figure 4, we plot the aggregate loss for two groups (high vs. low education) in the Default Credit dataset (Yeh & Lien, 2009) for different target dimensions $r$. By varying $t$, we achieve varying degrees of performance improvement on different groups—TERM ($t = 200$) recovers the min-max results of (Samadi et al., 2018) by forcing the losses on both groups to be (almost) identical, while TERM ($t = 10$) offers the flexibility of reducing the performance gap less aggressively.

**Handling class imbalance.** Next, we show that TERM can reduce the performance variance across classes with extremely imbalanced data when training deep neural networks. We compare TERM with several baselines which re-weight samples during training, including assigning weights inversely proportional to the class size (InverseRatio), focal loss (Lin et al., 2017), HardMine (Malisiewicz et al., 2011), and LearnReweight (Ren et al., 2018). Following (Ren et al., 2018), the datasets are composed of imbalanced 4 and 9 digits from MNIST (LeCun et al., 1998). In Figure 5, we see that TERM obtains similar (or higher) final accuracy on the clean test data as the state-of-the-art methods. We note that compared with LearnReweight, which optimizes the model over an additional balanced validation set and requires three gradient calculations for each update, TERM neither requires such balanced validation data nor does it increase the per-iteration complexity.

**Improving generalization via variance reduction.** A common alternative to ERM is to consider a distributionally robust objective, which optimizes for the worst-case training loss over a set of distributions, and has been shown to offer variance-reduction properties that benefit generalization (e.g., Namkoong & Duchi, 2017; Sinha et al., 2018). While not directly developed for distributional robustness, TERM also enables variance reduction for positive values of $t$ (Theorem 4), which can be used to strike a better bias-variance tradeoff for generalization. We compare TERM with several baselines including robustly regularized risk (RobustRegRisk) (Namkoong & Duchi, 2017), linear SVM (Ren et al., 2018), LearnReweight (Ren et al., 2018), FocalLoss (Lin et al., 2017), and HRM (Leqi et al., 2019). The results and detailed discussions are presented in Appendix I.2.

### 5.3 SOLVING COMPOUND ISSUES: HIERARCHICAL MULTI-OBJECTIVE TILTING

Finally, in this section, we focus on settings where multiple issues, e.g., class imbalance and label noise, exist in the data simultaneously. We discuss two possible instances of hierarchical multi-objective TERM to tackle such problems. One can think of other variants in this hierarchical tilting space which could be useful depending on applications at hand. However, we are not aware of other

prior work that aims to simultaneously handle multiple goals, e.g., suppressing noisy samples and addressing class imbalance, in a unified framework without additional validation data.

We explore the HIV-1 dataset (Rögnvaldsson, 2013), as in Section 5.2. We report both overall accuracy and accuracy on the rare class in four scenarios: **(a) clean and 1:4**, the original dataset that is naturally slightly imbalanced with rare samples represented 1:4 with respect to the common class; **(b) clean and 1:20**, where we subsample to introduce a 1:20 imbalance ratio; **(c) noisy and 1:4**, which is the original dataset with labels associated with 30% of the samples randomly reshuffled; and **(d) noisy and 1:20**, where 30% of the labels of the 1:20 imbalanced dataset are reshuffled.

Table 3: Hierarchical TERM can address both class imbalance and noisy samples.

| objectives | test accuracy (HIV-1) | | | | | | | |
|---|---|---|---|---|---|---|---|---|
| | clean data | | | | 30% noise | | | |
| | 1:4 | | 1:20 | | 1:4 | | 1:20 | |
| | $Y=0$ | overall | $Y=0$ | overall | $Y=0$ | overall | $Y=0$ | overall |
| ERM | 0.822 (.009) | 0.934 (.003) | 0.503 (.013) | 0.888 (.006) | 0.656 (.014) | 0.911 (.006) | 0.240 (.018) | 0.831 (.011) |
| GCE (Zhang & Sabuncu, 2018) | 0.822 (.009) | 0.934 (.003) | 0.503 (.013) | 0.888 (.006) | 0.732 (.021) | **0.925** (.005) | 0.324 (.017) | 0.849 (.008) |
| LearnReweight (Ren et al., 2018) | **0.841** (.014) | 0.934 (.004) | 0.800 (.022) | 0.904 (.003) | 0.721 (.034) | 0.856 (.008) | 0.532 (.054) | 0.856 (.013) |
| RobustRegRisk (Namkoong & Duchi, 2017) | **0.844** (.010) | **0.939** (.004) | 0.622 (.011) | 0.906 (.005) | 0.634 (.014) | 0.907 (.006) | 0.051 (.014) | 0.792 (.012) |
| FocalLoss (Lin et al., 2017) | 0.834 (.013) | **0.937** (.004) | 0.806 (.020) | **0.918** (.003) | 0.638 (.008) | 0.908 (.005) | 0.565 (.027) | **0.890** (.009) |
| TERM$_{sc}$ | 0.840 (.010) | **0.937** (.004) | **0.836** (.018) | **0.921** (.002) | **0.852** (.010) | 0.924 (.004) | **0.778** (.008) | **0.900** (.005) |
| TERM$_{ca}$ | **0.844** (.014) | **0.938** (.004) | **0.834** (.021) | **0.918** (.003) | **0.846** (.015) | **0.933** (.003) | **0.806** (.020) | **0.901** (.010) |

In Table 3, hierarchical TERM is applied at the sample level and class level (TERM$_{sc}$), where we use the sample-level tilt of $\tau=-2$ for noisy data. We use class-level tilt of $t=0.1$ for the 1:4 case and $t=50$ for the 1:20 case. We compare against baselines for robust classification and class imbalance (discussed previously in Sections 5.1 and 5.2), where we tune them for best performance (Appendix J). Similar to the experiments in Section 5.1, we avoid using baselines that require clean validation data (e.g., Roh et al., 2020). While different baselines perform well in their respective problem settings, TERM is far superior to all baselines when considering noisy samples and class imbalance simultaneously (rightmost column in Table 3). Finally, in the last row of Table 3, we simulate the noisy annotator setting of Section 5.1 assuming that the data is coming from 10 annotators, i.e., in the 30% noise case we have 7 hammers and 3 spammers. In this case, we apply hierarchical TERM at both class and annotator levels (TERM$_{ca}$), where we perform the higher level tilt at the annotator (group) level and the lower level tilt at the class level (with no sample-level tilting). We show that this approach can benefit noisy/imbalanced data even further (far right, Table 3), while suffering only a small performance drop on the clean and noiseless data (far left, Table 3).

## 6 RELATED WORK

**Alternate aggregation schemes: exponential smoothing/superquantile methods.** A common alternative to the standard average loss in empirical risk minimization is to consider a min-max objective, which aims to minimize the max-loss. Min-max objectives are commonplace in machine learning, and have been used for a wide range of applications, such as ensuring fairness across subgroups (Hashimoto et al., 2018; Mohri et al., 2019; Samadi et al., 2018; Stelmakh et al., 2019; Tantipongpipat et al., 2019), enabling robustness under small perturbations (Sinha et al., 2018), or generalizing to unseen domains (Volpi et al., 2018). As discussed in Section 2, the TERM objective can be viewed as a minimax smoothing (Kort & Bertsekas, 1972; Pee & Royset, 2011) with the added flexibility of a tunable $t$ to allow the user to optimize utility for different quantiles of loss similar to superquantile approaches (Laguel et al., 2021; Rockafellar et al., 2000), directly trading off between robustness/fairness and utility for positive and negative values of $t$ (see Appendix G for these connections). However, the TERM objective remains smooth (and efficiently solvable) for moderate values of $t$, resulting in faster convergence even when the resulting solutions are effectively the same as the min-max solution or other desired quantiles of the loss (as we demonstrate in the experiments of Section 5). Such smooth approximations to the max often appear through LogSumExp functions, with applications in geometric programming (Calafiore & El Ghaoui, 2014, Sec. 9.7), and boosting (Mason et al., 1999; Shen & Li, 2010). Interestingly, Cohen et al. introduce Simnets (Cohen & Shashua, 2014; Cohen et al., 2016), with a similar exponential smoothing operator, though for a differing purpose of achieving layer-wise operations *between* sum and max in deep neural networks.

**Alternate loss functions.** Rather than modifying the way the losses are aggregated, as in (smoothed) min-max or superquantile methods, it is also quite common to modify the losses themselves. For example, in robust regression, it is common to consider losses such as the $L_1$ loss, Huber loss, or general $M$-estimators (Holland & Ikeda, 2019) as a way to mitigate the effect of outliers (Bhatia et al., 2015). (Wang et al., 2013) studies a similar exponentially tilted loss for robust regression, though it is limited to the squared loss and only corresponds to $t<0$. Losses can also be modified to address outliers by favoring small losses (Yu et al., 2012; Zhang & Sabuncu, 2018) or gradient clipping (Menon et al., 2020). On the other extreme, the largest losses can be magnified to encourage focus on hard samples (Li et al., 2020b; Lin et al., 2017; Wang et al., 2016), which is a popular approach for curriculum learning. Constraints could also be imposed to promote fairness (Baharlouei et al., 2020; Donini et al., 2018; Hardt et al., 2016; Rezaei et al., 2020; Zafar et al., 2017). Ignoring the log portion of the objective in (2), TERM can be viewed as an alternate loss function exponentially shaping the loss to achieve both of these goals with a single objective, i.e., magnifying hard examples with $t > 0$ and suppressing outliers with $t < 0$. In addition, we show that TERM can even achieve both goals simultaneously with hierarchical multi-objective optimization (Section 5.3).

**Sample re-weighting schemes.** Finally, there exist approaches that implicitly modify the underlying ERM objective by re-weighting the influence of the samples themselves. These re-weighting schemes can be enforced in many ways. A simple and widely used example is to subsample training points in different classes. Alternatively, one can re-weight examples according to their loss function when using a stochastic optimizer, which can be used to put more emphasis on "hard" examples (Jiang et al., 2019; Katharopoulos & Fleuret, 2017; Leqi et al., 2019; Shrivastava et al., 2016). Re-weighting can also be implicitly enforced via the inclusion of a regularization parameter (Abdelkarim et al., 2020), loss clipping (Yang et al., 2010), or modelling crowd-worker qualities (Khetan et al., 2018). Such an explicit re-weighting has been explored for other applications (e.g., Chang et al., 2017; Gao et al., 2015; Jiang et al., 2018; Lin et al., 2017; Ren et al., 2018; Shu et al., 2019), though in contrast to these methods, TERM is applicable to a general class of loss functions, with theoretical guarantees. TERM is equivalent to a dynamic re-weighting of the samples based on the values of the objectives (Lemma 1), which could be viewed as a convexified version of loss clipping. We compare to several sample re-weighting schemes empirically in Section 5.

## 7    CONCLUSION

In this paper, we examined tilted empirical risk minimization (TERM) as a flexible extension to the ERM framework. We explored, both theoretically and empirically, TERM's ability to handle various known issues with ERM, such as robustness to noise, class imbalance, fairness, and generalization, as well as more complex issues like the simultaneous existence of class imbalance and noisy outliers. Despite the straightforward modification TERM makes to traditional ERM objectives, the framework consistently outperforms ERM and delivers competitive performance with state-of-the-art, problem-specific methods on a wide range of applications. Our work highlights the effectiveness and versatility of tilted objectives in machine learning. Building on the analyses and empirical study provided herein, in future work, it would be interesting to investigate generalization bounds for TERM as a function of $t$, and to derive theoretical convergence guarantees for our proposed stochastic solvers.

## ACKNOWLEDGEMENTS

We are grateful to Arun Sai Suggala and Adarsh Prasad (CMU) for their helpful comments on robust regression; to Zhiguang Wang, Dario Garcia Garcia, Alborz Geramifard, and other members of Facebook AI for productive discussions and feedback and pointers to prior work (Cohen & Shashua, 2014; Cohen et al., 2016; Rockafellar et al., 2000; Wang et al., 2016); and to Meisam Razaviyayn (USC) for helpful discussions and pointers to exponential smoothing (Kort & Bertsekas, 1972; Pee & Royset, 2011), Value-at-Risk (Nouiehed et al., 2019a; Rockafellar & Uryasev, 2002), and general properties of gradient-based methods in non-convex optimization problems (Ge et al., 2015; Jin et al., 2017; 2020; Ostrovskii et al., 2020). The work of TL and VS was supported in part by the National Science Foundation grant IIS1838017, a Google Faculty Award, a Carnegie Bosch Institute Research Award, and the CONIX Research Center. Any opinions, findings, and conclusions or recommendations expressed in this material are those of the author(s) and do not necessarily reflect the National Science Foundation or any other funding agency.

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

## APPENDIX

In this appendix we provide full statements and proofs of the analyses presented in Section 2 (Appendix A—G); details on the methods we propose for solving TERM (Appendix H); complete empirical results and details of our empirical setup (Appendix I—J), and a discussion on the broader impacts (both positive and negative) of TERM and the research herein (Appendix K). We provide a table of contents below for easier navigation.

## CONTENTS

## A  NOTATION & ASSUMPTIONS

In this section, we provide the notation and the assumptions that are used throughout our theoretical analyses.

The results in this paper are derived under one of the following four assumptions:

**Assumption 1** (Smoothness condition). *We assume that for $i \in [N]$, loss function $f(x_i; \theta)$ is in differentiability class $C^1$ (i.e., continuously differentiable) with respect to $\theta \in \Theta \subseteq \mathbb{R}^d$.*

**Assumption 2** (Strong convexity condition). *We assume that Assumption 1 is satisfied. In addition, we assume that for any $i \in [N]$, $f(x_i; \theta)$ is in differentiability class $C^2$ (i.e., twice differentiable with continuous Hessian) with respect to $\theta$. We further assume that there exist $\beta_{\min}, \beta_{\max} \in \mathbb{R}^+$ such that for $i \in [N]$ and any $\theta \in \Theta \subseteq \mathbb{R}^d$,*

$$\beta_{\min} \mathbf{I} \preceq \nabla^2_{\theta\theta^\top} f(x_i; \theta) \preceq \beta_{\max} \mathbf{I}, \tag{5}$$

*where $\mathbf{I}$ is the identity matrix of appropriate size (in this case $d \times d$). We further assume that there does **not** exist any $\theta \in \Theta$, such that $\nabla_\theta f(x_i; \theta) = 0$ for all $i \in [N]$.*

**Assumption 3** (Generalized linear model condition (Wainwright & Jordan, 2008)). *We assume that Assumption 2 is satisfied. We further assume that the loss function $f(x; \theta)$ is given by*

$$f(x; \theta) = A(\theta) - \theta^\top T(x), \tag{6}$$

*where $A(\cdot)$ is a convex function such that there exists $\beta_{\max}$ such that for any $\theta \in \Theta \subseteq \mathbb{R}^d$,*

$$\beta_{\min} \mathbf{I} \preceq \nabla^2_{\theta\theta^\top} A(\theta) \preceq \beta_{\max} \mathbf{I}. \tag{7}$$

*We also assume that*

$$\sum_{i \in [N]} T(x_i) T(x_i)^\top > 0. \tag{8}$$

This nest set of assumptions become the most restrictive with Assumption 3, which essentially requires that the loss be the negative log-likelihood of an exponential family. While the assumption is stated using the natural parameter of an exponential family for ease of presentation, the results hold for a bijective and smooth reparameterization of the exponential family. Assumption 3 is satisfied by the commonly used $L_2$ loss for regression and logistic loss for classification (see toy examples (b) and (c) in Figure 1). While the assumption is not satisfied when we use neural network function approximators in Section 5.1, we observe favorable numerical results motivating the extension of these results beyond the cases that are theoretically studied in this paper.

In the sequel, many of the results are concerned with characterizing the $t$-tilted solutions defined as the parametric set of solutions of $t$-tiled losses by sweeping $t \in \mathbb{R}$,

$$\breve{\theta}(t) \in \arg\min_{\theta \in \Theta} \widetilde{R}(t; \theta), \tag{9}$$

where $\Theta \subseteq \mathbb{R}^d$ is an open subset of $\mathbb{R}^d$. We state an assumption on this set below.

**Assumption 4** (Strict saddle property (Definition 4 in (Ge et al., 2015))). *We assume that the set $\arg\min_{\theta \in \Theta} \widetilde{R}(t; \theta)$ is non-empty for all $t \in \mathbb{R}$. Further, we assume that for all $t \in \mathbb{R}$, $\widetilde{R}(t; \theta)$ is a "strict saddle" as a function of $\theta$, i.e., for all local minima, $\nabla^2_{\theta\theta^\top} \widetilde{R}(t; \theta) > 0$, and for all other stationary solutions, $\lambda_{\min}(\nabla^2_{\theta\theta^\top} \widetilde{R}(t; \theta)) < 0$, where $\lambda_{\min}(\cdot)$ is the minimum eigenvalue of the matrix.*

We use the strict saddle property in order to reason about the properties of the $t$-tilted solutions. In particular, since we are solely interested in the local minima of $\widetilde{R}(t; \theta)$, the strict saddle property implies that for every $\breve{\theta}(t) \in \arg\min_{\theta \in \Theta} \widetilde{R}(t; \theta)$, for a sufficiently small $r$, for all $\theta \in \mathcal{B}(\breve{\theta}(t), r)$,

$$\nabla^2_{\theta\theta^\top} \widetilde{R}(t; \theta) > 0, \tag{10}$$

where $\mathcal{B}(\breve{\theta}(t), r)$ denotes a $d$-ball of radius $r$ around $\breve{\theta}(t)$.

We will show later that the strict saddle property is readily verified for $t \in \mathbb{R}^+$ under Assumption 2.

## B    BASIC PROPERTIES OF THE TERM OBJECTIVE

In this section, we provide the basic properties of the TERM objective.

**Proof of Lemma 1.** Lemma 1, which provides the gradient of the tilted objective, has been studied previously in the context of exponential smoothing (see (Pee & Royset, 2011, Proposition 2.1)). We provide a brief derivation here under Assumption 1 for completeness. We have:

$$\nabla_\theta \widetilde{R}(t;\theta) = \nabla_\theta \left\{ \frac{1}{t} \log \left( \frac{1}{N} \sum_{i\in[N]} e^{tf(x_i;\theta)} \right) \right\} \tag{11}$$

$$= \frac{\sum_{i\in[N]} \nabla_\theta f(x_i;\theta) e^{tf(x_i;\theta)}}{\sum_{i\in[N]} e^{tf(x_i;\theta)}} . \tag{12}$$

$\square$

**Lemma 2.** *Under Assumption 1,*

$$\widetilde{R}(-\infty;\theta) := \lim_{t\to-\infty} \widetilde{R}(t;\theta) = \check{R}(\theta), \tag{13}$$

$$\widetilde{R}(0;\theta) := \lim_{t\to0} \widetilde{R}(t;\theta) = \overline{R}(\theta), \tag{14}$$

$$\widetilde{R}(+\infty;\theta) := \lim_{t\to+\infty} \widetilde{R}(t;\theta) = \widehat{R}(\theta), \tag{15}$$

*where $\widehat{R}(\theta)$ is the max-loss and $\check{R}(\theta)$ is the min-loss[2]:*

$$\widehat{R}(\theta) := \max_{i\in[N]} f(x_i;\theta), \qquad\qquad \check{R}(\theta) := \min_{i\in[N]} f(x_i;\theta). \tag{16}$$

*Proof.* For $t \to 0$,

$$\lim_{t\to0} \widetilde{R}(t;\theta) = \lim_{t\to0} \frac{1}{t} \log \left( \frac{1}{N} \sum_{i\in[N]} e^{tf(x_i;\theta)} \right)$$

$$= \lim_{t\to0} \frac{\sum_{i\in[N]} f(x_i;\theta) e^{tf(x_i;\theta)}}{\sum_{i\in[N]} e^{tf(x_i;\theta)}} \tag{17}$$

$$= \frac{1}{N} \sum_{i\in[N]} f(x_i;\theta), \tag{18}$$

where (17) is due to L'Hôpital's rule applied to $t$ as the denominator and $\log\left(\frac{1}{N}\sum_{i\in[N]} e^{tf(x_i;\theta)}\right)$ as the numerator.

For $t \to -\infty$, we proceed as follows:

$$\lim_{t\to-\infty} \widetilde{R}(t;\theta) = \lim_{t\to-\infty} \frac{1}{t} \log \left( \frac{1}{N} \sum_{i\in[N]} e^{tf(x_i;\theta)} \right)$$

$$\geqslant \lim_{t\to-\infty} \frac{1}{t} \log \left( \frac{1}{N} \sum_{i\in[N]} e^{t\min_{j\in[N]} f(x_j;\theta)} \right) \tag{19}$$

$$= \min_{i\in[N]} f(x_i;\theta). \tag{20}$$

---

[2]When the argument of the max-loss or the min-loss is not unique, for the purpose of differentiating the loss function, we define $\widehat{R}(\theta)$ as the average of the individual losses that achieve the maximum, and $\check{R}(\theta)$ as the average of the individual losses that achieve the minimum.

On the other hand,

$$\lim_{t \to -\infty} \widetilde{R}(t; \theta) = \lim_{t \to -\infty} \frac{1}{t} \log \left( \frac{1}{N} \sum_{i \in [N]} e^{tf(x_i; \theta)} \right)$$

$$\leqslant \lim_{t \to -\infty} \frac{1}{t} \log \left( \frac{1}{N} e^{t \min_{j \in [N]} f(x_j; \theta)} \right) \tag{21}$$

$$= \min_{i \in [N]} f(x_i; \theta) - \lim_{t \to -\infty} \left\{ \frac{1}{t} \log N \right\} \tag{22}$$

$$= \min_{i \in [N]} f(x_i; \theta). \tag{23}$$

Hence, the proof follows by putting together (20) and (23).

The proof proceeds similarly to $t \to -\infty$ for $t \to +\infty$ and is omitted for brevity. $\qquad \square$

Note that Lemma 2 has been previously observed in (Cohen & Shashua, 2014). This lemma also implies that $\widetilde{\theta}(0)$ is the ERM solution, $\widetilde{\theta}(+\infty)$ is the min-max solution, and $\widetilde{\theta}(-\infty)$ is the min-min solution.

**Lemma 3** (Tilted Hessian and strong convexity for $t \in \mathbb{R}^+$). *Under Assumption 2, for any $t \in \mathbb{R}$,*

$$\nabla^2_{\theta\theta^\top} \widetilde{R}(t; \theta) = t \sum_{i \in [N]} (\nabla_\theta f(x_i; \theta) - \nabla_\theta \widetilde{R}(t; \theta))(\nabla_\theta f(x_i; \theta) - \nabla_\theta \widetilde{R}(t; \theta))^\top e^{t(f(x_i; \theta) - \widetilde{R}(t; \theta))}$$

$$\tag{24}$$

$$+ \sum_{i \in [N]} \nabla^2_{\theta\theta^\top} f(x_i; \theta) e^{t(f(x_i; \theta) - \widetilde{R}(t; \theta))}. \tag{25}$$

*In particular, for all $\theta \in \Theta$ and all $t \in \mathbb{R}^+$, the $t$-tilted objective is strongly convex. That is*

$$\nabla^2_{\theta\theta^\top} \widetilde{R}(t; \theta) > \beta_{\min} \mathbf{I}. \tag{26}$$

*Proof.* Recall that

$$\nabla_\theta \widetilde{R}(t; \theta) = \frac{\sum_{i \in [N]} \nabla_\theta f(x_i; \theta) e^{tf(x_i; \theta)}}{\sum_{i \in [N]} e^{tf(x_i; \theta)}} \tag{27}$$

$$= \sum_{i \in [N]} \nabla_\theta f(x_i; \theta) e^{t(f(x_i; \theta) - \widetilde{R}(t; \theta))}. \tag{28}$$

The proof of the first part is completed by differentiating again with respect to $\theta$, followed by algebraic manipulation. To prove the second part, notice that the term in (24) is positive semi-definite, whereas the term in (25) is positive definite and lower bounded by $\beta_{\min} \mathbf{I}$ (see Assumption 2, Eq. (5)). $\qquad \square$

**Lemma 4** (Smoothness of $\widetilde{R}(t; \theta)$ in the vicinity of the final solution $\breve{\theta}(t)$). *For any $t \in \mathbb{R}$, let $\beta(t)$ be the smoothness parameter in the vicinity of the final solution:*

$$\beta(t) := \sup_{\theta \in \mathcal{B}(\breve{\theta}(t), r)} \lambda_{\max} \left( \nabla^2_{\theta\theta^\top} \widetilde{R}(t; \theta) \right), \tag{29}$$

*where $\nabla^2_{\theta\theta^\top} \widetilde{R}(t; \theta)$ is the Hessian of $\widetilde{R}(t; \theta)$ at $\theta$, $\lambda_{\max}(\cdot)$ denotes the largest eigenvalue, and $\mathcal{B}(\theta, r)$ denotes a $d$-ball of radius $r$ around $\theta$. Under Assumption 2, for any $t \in \mathbb{R}$, $\widetilde{R}(t; \theta)$ is a $\beta(t)$-smooth function of $\theta$. Further, for $t \in \mathbb{R}^-$, at the vicinity of $\breve{\theta}(t)$,*

$$\beta(t) < \beta_{\max}, \tag{30}$$

*and for $t \in \mathbb{R}^+$,*

$$0 < \lim_{t \to +\infty} \frac{\beta(t)}{t} < +\infty. \tag{31}$$

*Proof.* Let us first provide a proof for $t \in \mathbb{R}^-$. Invoking Lemma 3 and Weyl's inequality (Weyl, 1912), we have

$$\lambda_{\max} \left( \nabla^2_{\theta\theta^\top} \widetilde{R}(t;\theta) \right)$$

$$\leqslant \lambda_{\max} \left( t \sum_{i \in [N]} (\nabla_\theta f(x_i;\theta) - \nabla_\theta \widetilde{R}(t;\theta))(\nabla_\theta f(x_i;\theta) - \nabla_\theta \widetilde{R}(t;\theta))^\top e^{t(f(x_i;\theta) - \widetilde{R}(t;\theta))} \right) \tag{32}$$

$$+ \lambda_{\max} \left( \sum_{i \in [N]} \nabla^2_{\theta\theta^\top} f(x_i;\theta) e^{t(f(x_i;\theta) - \widetilde{R}(t;\theta))} \right) \tag{33}$$

$$\leqslant \beta_{\max}, \tag{34}$$

where we have used the fact that the term in (24) is negative semi-definite for $t < 0$, and that the term in (25) is positive definite for all $t$ with smoothness bounded by $\beta_{\max}$ (see Assumption 2, Eq. (5)).

For $t \in \mathbb{R}^+$, following Lemma 3 and Weyl's inequality (Weyl, 1912), we have

$$\left( \frac{1}{t} \right) \lambda_{\max} \left( \nabla^2_{\theta\theta^\top} \widetilde{R}(t;\theta) \right)$$

$$\leqslant \lambda_{\max} \left( \sum_{i \in [N]} (\nabla_\theta f(x_i;\theta) - \nabla_\theta \widetilde{R}(t;\theta))(\nabla_\theta f(x_i;\theta) - \nabla_\theta \widetilde{R}(t;\theta))^\top e^{t(f(x_i;\theta) - \widetilde{R}(t;\theta))} \right) \tag{35}$$

$$+ \left( \frac{1}{t} \right) \lambda_{\max} \left( \sum_{i \in [N]} \nabla^2_{\theta\theta^\top} f(x_i;\theta) e^{t(f(x_i;\theta) - \widetilde{R}(t;\theta))} \right). \tag{36}$$

Consequently,

$$\lim_{t \to +\infty} \left( \frac{1}{t} \right) \lambda_{\max} \left( \nabla^2_{\theta\theta^\top} \widetilde{R}(t;\theta) \right) < +\infty. \tag{37}$$

On the other hand, following Weyl's inequality (Weyl, 1912),

$$\lambda_{\max} \left( \nabla^2_{\theta\theta^\top} \widetilde{R}(t;\theta) \right)$$

$$\geqslant t \lambda_{\max} \left( \sum_{i \in [N]} (\nabla_\theta f(x_i;\theta) - \nabla_\theta \widetilde{R}(t;\theta))(\nabla_\theta f(x_i;\theta) - \nabla_\theta \widetilde{R}(t;\theta))^\top e^{t(f(x_i;\theta) - \widetilde{R}(t;\theta))} \right), \tag{38}$$

and hence,

$$\lim_{t \to +\infty} \left( \frac{1}{t} \right) \lambda_{\max} \left( \nabla^2_{\theta\theta^\top} \widetilde{R}(t;\theta) \right) > 0, \tag{39}$$

where we have used the fact that no solution $\theta$ exists that would make all $f_i$'s vanish (Assumption 2). $\square$

Under the strict saddle property (Assumption 4), it is known that gradient-based methods would converge to a local minimum (Ge et al., 2015), i.e., $\breve{\theta}(t)$ would be obtained using gradient descent (GD). The rate of convergence of GD scales linearly with the smoothness parameter of the optimization landscape, which is characterized by Lemma 4.

**Lemma 5** (Strong convexity of $\widetilde{R}(t;\theta)$ in $\mathbb{R}^+$). *Under Assumption 2, for any $t \in \mathbb{R}^+$, $\widetilde{R}(t;\theta)$ is a strongly convex function of $\theta$. That is for $t \in \mathbb{R}^+$,*

$$\nabla^2_{\theta\theta^\top} \widetilde{R}(t;\theta) > \beta_{\min} \mathbf{I}. \tag{40}$$

*Proof.* The result follows by invoking Lemma 3 with $t \in \mathbb{R}^+$, and considering (5) (Assumption 2). $\square$

This lemma also implies that under Assumption 2, the strict saddle assumption (Assumption 4) is readily verified.

## C   Hierarchical Multi-Objective Tilting

We start by stating the hierarchical multi-objective tilting for a hierarchy of depth 3. While we don't directly use this form, it is stated to clarify the experiments in Section 5 where tilting is done at class level and annotator level, and the sample-level tilt value could be understood to be 0.

$$\widetilde{J}(m, t, \tau; \theta) := \frac{1}{m} \log \left( \frac{1}{N} \sum_{G \in [GG]} \left( \sum_{g \in [G]} |g| \right) e^{m \widetilde{J}_G(\tau; \theta)} \right) \tag{41}$$

$$\widetilde{J}_G(t, \tau; \theta) := \frac{1}{t} \log \left( \frac{1}{\sum_{g \in [G]} |g|} \sum_{g \in [G]} |g| e^{t \widetilde{R}_g(\tau; \theta)} \right) \tag{42}$$

$$\widetilde{R}_g(\tau; \theta) := \frac{1}{\tau} \log \left( \frac{1}{|g|} \sum_{x \in g} e^{\tau f(x; \theta)} \right), \tag{43}$$

Next, we continue by evaluating the gradient of the hierarchical multi-objective tilt for a hierarchy of depth 2.

**Lemma 6** (Hierarchical multi-objective tilted gradient). *Under Assumption 1,*

$$\nabla_\theta \widetilde{J}(t, \tau; \theta) = \sum_{g \in [G]} \sum_{x \in g} w_{g,x}(t, \tau; \theta) \nabla_\theta f(x; \theta) \tag{44}$$

*where*

$$w_{g,x}(t, \tau; \theta) := \frac{\left( \frac{1}{|g|} \sum_{y \in g} e^{\tau f(y; \theta)} \right)^{\left( \frac{t}{\tau} - 1 \right)}}{\sum_{g' \in [G]} |g'| \left( \frac{1}{|g'|} \sum_{y \in g'} e^{\tau f(y; \theta)} \right)^{\frac{t}{\tau}}} \; e^{\tau f(x; \theta)}. \tag{45}$$

*Proof.* We proceed as follows. First notice that by invoking Lemma 1,

$$\nabla_\theta \widetilde{J}(t, \tau; \theta) = \sum_{g \in [G]} w_g(t, \tau; \theta) \nabla_\theta \widetilde{R}_g(\tau; \theta) \tag{46}$$

where

$$w_g(t, \tau; \theta) := \frac{|g| e^{t \widetilde{R}_g(\tau; \theta)}}{\sum_{g' \in [G]} |g'| e^{t \widetilde{R}_{g'}(\tau; \theta)}}. \tag{47}$$

where $\widetilde{R}_g(\tau; \theta)$ is defined in (4), and is reproduced here:

$$\widetilde{R}_g(\tau; \theta) := \frac{1}{\tau} \log \left( \frac{1}{|g|} \sum_{x \in g} e^{\tau f(x; \theta)} \right). \tag{48}$$

On the other hand, by invoking Lemma 1,

$$\nabla_\theta \widetilde{R}_g(\tau; \theta) = \sum_{x \in g} w_{g,x}(\tau; \theta) \nabla_\theta f(x; \theta) \tag{49}$$

where

$$w_{g,x}(\tau; \theta) := \frac{e^{\tau f(x; \theta)}}{\sum_{y \in g} e^{\tau f(y; \theta)}}. \tag{50}$$

Hence, combining (46) and (49),

$$\nabla_\theta \widetilde{J}(t, \tau; \theta) = \sum_{g \in [G]} \sum_{x \in g} w_g(t, \tau; \theta) w_{g,x}(\tau; \theta) \nabla_\theta f(x; \theta). \tag{51}$$

The proof is completed by algebraic manipulations to show that

$$w_{g,x}(t, \tau; \theta) = w_g(t, \tau; \theta) w_{g,x}(\tau; \theta). \tag{52}$$

$\square$

**Lemma 7** (Sample-level TERM is a special case of hierarchical multi-objective TERM). *Under Assumption 1, hierarchical multi-objective TERM recovers TERM as a special case for $t = \tau$. That is*

$$\widetilde{J}(t, t; \theta) = \widetilde{R}(t; \theta). \tag{53}$$

*Proof.* The proof is completed by noticing that setting $t = \tau$ in (45) (Lemma 6) recovers the original sample-level tilted gradient. □

## D    GENERAL PROPERTIES OF THE OBJECTIVE FOR GLMs

In this section, even if not explicitly stated, all results are derived under Assumption 3 with a generalized linear model and loss function of the form (6), effectively assuming that the loss function is the negative log-likelihood of an exponential family (Wainwright & Jordan, 2008).

**Definition 1** (Empirical cumulant generating function). *Let*

$$\Lambda(t;\theta) := t\widetilde{R}(t;\theta). \tag{54}$$

**Definition 2** (Empirical log-partition function (Wainwright et al., 2005)). *Let* $\Gamma(t;\theta)$ *be*

$$\Gamma(t;\theta) := \log\left(\frac{1}{N}\sum_{i\in[N]} e^{-t\theta^\top T(x_i)}\right). \tag{55}$$

Thus, we have

$$\widetilde{R}(t;\theta) = A(\theta) + \frac{1}{t}\log\left(\frac{1}{N}\sum_{i\in[N]} e^{-t\theta^\top T(x_i)}\right) = A(\theta) + \frac{1}{t}\Gamma(t;\theta). \tag{56}$$

**Definition 3** (Empirical mean and empirical variance of the sufficient statistic). *Let* $\mathcal{M}$ *and* $\mathcal{V}$ *denote the mean and the variance of the sufficient statistic, and be given by*

$$\mathcal{M}(t;\theta) := \frac{1}{N}\sum_{i\in[N]} T(x_i)e^{-t\theta^\top T(x_i)-\Gamma(t;\theta)}, \tag{57}$$

$$\mathcal{V}(t;\theta) := \frac{1}{N}\sum_{i\in[N]} (T(x_i)-\mathcal{M}(t;\theta))(T(x_i)-\mathcal{M}(t;\theta))^\top e^{-t\theta^\top T(x_i)-\Gamma(t;\theta)}. \tag{58}$$

**Lemma 8.** *For all* $t \in \mathbb{R}$, *we have* $\mathcal{V}(t;\theta) > 0$.

Next we state a few key relationships that we will use in our characterizations. The proofs are straightforward and omitted for brevity.

**Lemma 9** (Partial derivatives of $\Gamma$). *For all* $t \in \mathbb{R}$ *and all* $\theta \in \Theta$,

$$\frac{\partial}{\partial t}\Gamma(t;\theta) = -\theta^\top \mathcal{M}(t;\theta), \tag{59}$$

$$\nabla_\theta \Gamma(t;\theta) = -t\mathcal{M}(t;\theta). \tag{60}$$

**Lemma 10** (Partial derivatives of $\mathcal{M}$). *For all* $t \in \mathbb{R}$ *and all* $\theta \in \Theta$,

$$\frac{\partial}{\partial t}\mathcal{M}(t;\theta) = -\mathcal{V}(t;\theta)\theta, \tag{61}$$

$$\nabla_\theta \mathcal{M}(t;\theta) = -t\mathcal{V}(t;\theta). \tag{62}$$

The next few lemmas characterize the partial derivatives of the cumulant generating function.

**Lemma 11.** *(Derivative of $\Lambda$ with t)* *For all* $t \in \mathbb{R}$ *and all* $\theta \in \Theta$,

$$\frac{\partial}{\partial t}\Lambda(t;\theta) = A(\theta) - \theta^\top \mathcal{M}(t;\theta). \tag{63}$$

*Proof.* The proof is carried out by

$$\frac{\partial}{\partial t}\Lambda(t;\theta) = A(\theta) - \theta^\top \sum_{i\in[N]} T(x_i)e^{-t\theta^\top T(x_i)-\Gamma(t;\theta)} = A(\theta) - \theta^\top \mathcal{M}(t;\theta). \tag{64}$$

$\square$

**Lemma 12** (Second derivative of $\Lambda$ with $t$). *For all* $t \in \mathbb{R}$ *and all* $\theta \in \Theta$,

$$\frac{\partial^2}{\partial t^2}\Lambda(t;\theta) = \theta^\top \mathcal{V}(t;\theta)\theta. \tag{65}$$

**Lemma 13** (Gradient of $\Lambda$ with $\theta$). *For all $t \in \mathbb{R}$ and all $\theta \in \Theta$,*

$$\nabla_\theta \Lambda(t; \theta) = t \nabla_\theta A(\theta) - t \mathcal{M}(t; \theta). \tag{66}$$

**Lemma 14** (Hessian of $\Lambda$ with $\theta$). *For all $t \in \mathbb{R}$ and all $\theta \in \Theta$,*

$$\nabla^2_{\theta\theta^\top} \Lambda(t; \theta) = t \nabla^2_{\theta\theta^\top} A(\theta) + t^2 \mathcal{V}(t; \theta). \tag{67}$$

**Lemma 15** (Gradient of $\Lambda$ with respect to $t$ and $\theta$). *For all $t \in \mathbb{R}$ and all $\theta \in \Theta$,*

$$\frac{\partial}{\partial t} \nabla_\theta \Lambda(t; \theta) = \nabla_\theta A(\theta) - \mathcal{M}(t; \theta) + t \mathcal{V}(t; \theta) \theta. \tag{68}$$

# E    GENERAL PROPERTIES OF TERM SOLUTIONS FOR GLMS

Next, we characterize some of the general properties of the solutions of TERM objectives. Note that these properties are established under Assumptions 3 and 4.

**Lemma 16.** *For all $t \in \mathbb{R}$,*

$$\nabla_\theta \Lambda(t; \breve\theta(t)) = 0. \tag{69}$$

*Proof.* The proof follows from definition and the assumption that $\Theta$ is an open set. $\qquad\square$

**Lemma 17.** *For all $t \in \mathbb{R}$,*

$$\nabla_\theta A(\breve\theta(t)) = \mathcal{M}(t; \breve\theta(t)). \tag{70}$$

*Proof.* The proof is completed by noting Lemma 16 and Lemma 13. $\qquad\square$

**Lemma 18** (Derivative of the solution with respect to tilt)**.** *Under Assumption 4, for all $t \in \mathbb{R}$,*

$$\frac{\partial}{\partial t}\breve\theta(t) = -\left(\nabla^2_{\theta\theta^\top} A(\breve\theta(t)) + t\mathcal{V}(t; \breve\theta(t))\right)^{-1} \mathcal{V}(t; \breve\theta(t))\breve\theta(t), \tag{71}$$

*where*

$$\nabla^2_{\theta\theta^\top} A(\breve\theta(t)) + t\mathcal{V}(t; \breve\theta(t)) > 0. \tag{72}$$

*Proof.* By noting Lemma 16, and further differentiating with respect to $t$, we have

$$0 = \frac{\partial}{\partial t}\nabla_\theta \Lambda(t; \breve\theta(t)) \tag{73}$$

$$= \frac{\partial}{\partial \tau}\nabla_\theta \Lambda(\tau; \breve\theta(t))\bigg|_{\tau=t} + \nabla^2_{\theta\theta^\top}\Lambda(t; \breve\theta(t))\left(\frac{\partial}{\partial t}\breve\theta(t)\right) \tag{74}$$

$$= t\mathcal{V}(t; \breve\theta(t))\breve\theta(t) + \left(t\nabla^2_{\theta\theta^\top} A(\theta) + t^2\mathcal{V}(t; \theta)\right)\left(\frac{\partial}{\partial t}\breve\theta(t)\right), \tag{75}$$

where (74) follows from the chain rule, (75) follows from Lemmas 15 and 17 and 14. The proof is completed by noting that $\nabla^2_{\theta\theta^\top}\Lambda(t; \breve\theta(t)) > 0$ for all $t \in \mathbb{R}$ under Assumption 4. $\qquad\square$

Finally, we state an auxiliary lemma that will be used in the proof of the main theorem.

**Lemma 19.** *For all $t, \tau \in \mathbb{R}$ and all $\theta \in \Theta$,*

$$\mathcal{M}(\tau; \theta) - \mathcal{M}(t; \theta) = -\left(\int_t^\tau \mathcal{V}(\nu; \theta)d\nu\right)\theta. \tag{76}$$

*Proof.* The proof is completed by noting that

$$\mathcal{M}(\tau; \theta) - \mathcal{M}(t; \theta) = \int_t^\tau \frac{\partial}{\partial \nu}\mathcal{M}(\nu; \theta)d\nu = -\left(\int_t^\tau \mathcal{V}(\nu; \theta)d\nu\right)\theta. \tag{77}$$

$\qquad\square$

**Theorem 1.** *Under Assumption 3 and Assumption 4, for any $t, \tau \in \mathbb{R}$,*
*(a) $\frac{\partial}{\partial t}\widetilde{R}(\tau; \breve\theta(t)) < 0$ iff $t < \tau$;    (b) $\frac{\partial}{\partial t}\widetilde{R}(\tau; \breve\theta(t)) = 0$ iff $t = \tau$;    (c) $\frac{\partial}{\partial t}\widetilde{R}(\tau; \breve\theta(t)) > 0$ iff $t > \tau$.*

*Proof.* The proof proceeds as follows. Notice that

$$\frac{\partial}{\partial \tau}\widetilde{R}(t;\breve{\theta}(\tau)) = \frac{1}{t}\left(\frac{\partial}{\partial \tau}\breve{\theta}(\tau)\right)^{\top}\nabla_{\theta}\Lambda(t;\breve{\theta}(\tau)) \tag{78}$$

$$= -\breve{\theta}^{\top}(\tau)\mathcal{V}(\tau;\breve{\theta}(\tau))\left(\nabla^2_{\theta\theta^{\top}}A(\breve{\theta}(\tau)) + \tau\mathcal{V}(\tau;\breve{\theta}(\tau))\right)^{-1}$$
$$\times \left(\nabla_{\theta}A(\breve{\theta}(\tau)) - \mathcal{M}(t;\breve{\theta}(\tau))\right) \tag{79}$$

$$= -\breve{\theta}^{\top}(\tau)\mathcal{V}(\tau;\breve{\theta}(\tau))\left(\nabla^2_{\theta\theta^{\top}}A(\breve{\theta}(\tau)) + \tau\mathcal{V}(\tau;\breve{\theta}(\tau))\right)^{-1}$$
$$\times \left(\mathcal{M}(\tau;\breve{\theta}(\tau)) - \mathcal{M}(t;\breve{\theta}(\tau))\right) \tag{80}$$

$$= \breve{\theta}^{\top}(\tau)\mathcal{V}(\tau;\breve{\theta}(\tau))\left(\nabla^2_{\theta\theta^{\top}}A(\breve{\theta}(\tau)) + \tau\mathcal{V}(\tau;\breve{\theta}(\tau))\right)^{-1}$$
$$\times \left(\int_t^{\tau}\mathcal{V}(\nu;\breve{\theta}(\tau))d\nu\right)\breve{\theta}(\tau), \tag{81}$$

where (78) follows from the chain rule and (54), (79) follows from Lemma 18 and Lemma 13, (80) follows from Lemma 17, and (81) follows from Lemma 19. Now notice that invoking Lemma 8, and noticing that following the strict saddle property

$$\nabla^2_{\theta\theta^{\top}}\widetilde{R}(t;\theta)\Big|_{\theta=\breve{\theta}(\tau)} = \nabla^2_{\theta\theta^{\top}}A(\breve{\theta}(\tau)) + \tau\mathcal{V}(\tau;\breve{\theta}(\tau)) > 0, \tag{82}$$

we have

(a) $\int_t^{\tau}\mathcal{V}(\nu;\breve{\theta}(\tau))d\nu < 0$ iff $t < \tau$;

(b) $\int_t^{\tau}\mathcal{V}(\nu;\breve{\theta}(\tau))d\nu = 0$ iff $t = \tau$;

(c) $\int_t^{\tau}\mathcal{V}(\nu;\breve{\theta}(\tau))d\nu > 0$ iff $t > \tau$,

which completes the proof. $\qquad\square$

**Theorem 2** (Average- vs. max-loss tradeoff). *Under Assumption 3 and Assumption 4, for any $t \in \mathbb{R}^{+}$,*

$$\frac{\partial}{\partial t}\widehat{R}(\breve{\theta}(t)) \leqslant 0, \tag{83}$$

$$\frac{\partial}{\partial t}\overline{R}(\breve{\theta}(t)) \geqslant 0. \tag{84}$$

*Proof of Theorem 2.* To prove (83), first notice that from Lemma 2,

$$\widehat{R}(\theta) = \lim_{t\to+\infty}\widetilde{R}(t;\theta). \tag{85}$$

Now, invoking Theorem 1 (Appendix D), for any $\tau, t \in \mathbb{R}^{+}$ such that $\tau < t$

$$\frac{\partial}{\partial \tau}\widetilde{R}(t;\breve{\theta}(\tau)) < 0, \tag{86}$$

In particular, by taking the limit as $t \to +\infty$,

$$\lim_{t\to+\infty}\frac{\partial}{\partial \tau}\widetilde{R}(t;\breve{\theta}(\tau)) \leqslant 0. \tag{87}$$

Notice that

$$0 \geqslant \lim_{t \to +\infty} \frac{\partial}{\partial \tau} \widetilde{R}(t; \breve{\theta}(\tau)) = \lim_{t \to +\infty} \left( \frac{\partial}{\partial \tau} \breve{\theta}(\tau) \right)^{\top} \nabla_{\theta} \widetilde{R}(t; \breve{\theta}(\tau)) \tag{88}$$

$$= \left( \frac{\partial}{\partial \tau} \breve{\theta}(\tau) \right)^{\top} \lim_{t \to +\infty} \nabla_{\theta} \widetilde{R}(t; \breve{\theta}(\tau)) \tag{89}$$

$$= \left( \frac{\partial}{\partial \tau} \breve{\theta}(\tau) \right)^{\top} \nabla_{\theta} \widehat{R}(\breve{\theta}(\tau)) \tag{90}$$

$$= \frac{\partial}{\partial \tau} \widehat{R}(\breve{\theta}(\tau)) \tag{91}$$

where (90) holds because $\nabla_{\theta} \widetilde{R}(t; \breve{\theta}(\tau))$ is a finite weighted sum of the gradients of the individual losses with weights bounded in $[0, 1]$, per Lemma 1, completing the proof of the first part.

To prove (84), notice that by Lemma 2,

$$\overline{R}(\theta) = \lim_{t \to 0} \widetilde{R}(t; \theta). \tag{92}$$

Now, invoking Theorem 1 (Appendix D), for any $\tau, t \in \mathbb{R}^+$ such that $\tau > t$

$$\frac{\partial}{\partial \tau} \widetilde{R}(t; \breve{\theta}(\tau)) > 0. \tag{93}$$

In particular, by taking the limit as $t \to 0$,

$$\frac{\partial}{\partial \tau} \overline{R}(\breve{\theta}(\tau)) = \lim_{t \to 0} \frac{\partial}{\partial \tau} \widetilde{R}(t; \breve{\theta}(\tau)) > 0, \tag{94}$$

completing the proof. $\qquad \square$

**Theorem 3** (Average- vs. min-loss tradeoff). *Under Assumption 3 and Assumption 4, for any $t \in \mathbb{R}^-$,*

$$\frac{\partial}{\partial t} \breve{R}(\widetilde{\theta}(t)) \geqslant 0, \tag{95}$$

$$\frac{\partial}{\partial t} \overline{R}(\widetilde{\theta}(t)) \leqslant 0. \tag{96}$$

*Proof of Theorem 3.* To prove (95), first notice that from Lemma 2,

$$\widehat{R}(\theta) = \lim_{t \to -\infty} \widetilde{R}(t; \theta). \tag{97}$$

Now, invoking Theorem 1 (Appendix D), for any $\tau, t \in \mathbb{R}^+$ such that $\tau > t$

$$\frac{\partial}{\partial \tau} \widetilde{R}(t; \breve{\theta}(\tau)) > 0. \tag{98}$$

In particular, by taking the limit as $t \to -\infty$,

$$\frac{\partial}{\partial \tau} \breve{R}(\breve{\theta}(\tau)) = \lim_{t \to -\infty} \frac{\partial}{\partial \tau} \widetilde{R}(t; \breve{\theta}(\tau)) > 0, \tag{99}$$

completing the proof of the first part.

To prove (96), notice that by Lemma 2,

$$\overline{R}(\theta) = \lim_{t \to 0} \widetilde{R}(t; \theta). \tag{100}$$

Now, invoking Theorem 1 (Appendix D), for any $\tau, t \in \mathbb{R}^+$ such that $\tau < t$

$$\frac{\partial}{\partial \tau} \widetilde{R}(t; \breve{\theta}(\tau)) < 0. \tag{101}$$

In particular, by taking the limit as $t \to 0$,

$$\frac{\partial}{\partial \tau} \overline{R}(\breve{\theta}(\tau)) = \lim_{t \to 0} \frac{\partial}{\partial \tau} \widetilde{R}(t; \breve{\theta}(\tau)) < 0, \tag{102}$$

completing the proof. $\qquad \square$

Theorem 1 is concerned with characterizing the impact that TERM solutions for different $t \in \mathbb{R}$ have on the objective $\widetilde{R}(\tau; \breve{\theta}(t))$ for some fixed $\tau \in \mathbb{R}$. Recall that $\tau = -\infty$ recovers the min-loss, $\tau = 0$ is the average-loss, and $\tau = +\infty$ is the max-loss. By definition, if $t = \tau$, $\breve{\theta}(\tau)$ is the minimizer of $\widetilde{R}(\tau; \breve{\theta}(t))$. Theorem 1 shows that for $t \in (-\infty, \tau)$ the objective is *decreasing*; while for $t \in (\tau, +\infty)$ the objective *increasing*. Recall that for any fixed $\tau \in \mathbb{R}$, $\widetilde{R}(\tau; \theta)$ is also related to the $k$-th smallest loss of the population (Appendix G). Hence, the solution $\breve{\theta}(t)$ is approximately minimizing the $k(t)$-th smallest loss where $k(t)$ is increasing from 1 to $N$ by sweeping $t$ in $(-\infty, +\infty)$.

**Theorem 4** (Variance reduction). *Let* $\mathbf{f}(\theta) := (f(x_1; \theta)), \ldots, f(x_N; \theta))$. *For any* $\mathbf{u} \in \mathbf{R}^N$, *let*

$$\text{mean}(\mathbf{u}) := \frac{1}{N} \sum_{i \in [N]} u_i, \qquad \text{var}(\mathbf{u}) := \frac{1}{N} \sum_{i \in [N]} (u_i - \text{mean}(\mathbf{u}))^2. \tag{103}$$

*Then, under Assumption 3 and Assumption 4, for any* $t \in \mathbb{R}$,

$$\frac{\partial}{\partial t} \left\{ \text{var}(\mathbf{f}(\breve{\theta}(t))) \right\} < 0. \tag{104}$$

*Proof.* Recall that $f(x_i; \theta) = A(\theta) - \theta^\top T(x_i)$. Thus,

$$\text{mean}(\mathbf{f}) = \frac{1}{N} \sum_{i \in [N]} f(x_i; \theta) = A(\theta) - \frac{1}{N} \theta^\top \sum_{i \in [N]} T(x_i) = A(\theta) - \mathcal{M}(0; \theta) \tag{105}$$

Consequently,

$$\text{var}(\mathbf{f}(\theta)) = \frac{1}{N} \sum_{i \in [N]} \left( f(x_i; \theta) - \frac{1}{N} \sum_{j \in [N]} f(x_j; \theta) \right)^2 \tag{106}$$

$$= \frac{1}{N} \sum_{i \in [N]} \left( \theta^\top T(x_i) - \frac{1}{N} \theta^\top \sum_{j \in [N]} T(x_j) \right)^2 \tag{107}$$

$$= \frac{1}{N} \theta^\top \left( \sum_{i \in [N]} (T(x_i) - \frac{1}{N} \sum_{j \in [N]} T(x_j))(T(x_i) - \frac{1}{N} \sum_{j \in [N]} T(x_j))^\top \right) \theta \tag{108}$$

$$= \theta^\top \mathcal{V}_0 \theta, \tag{109}$$

where

$$\mathcal{V}_0 = \mathcal{V}(0; \theta) = \frac{1}{N} \sum_{i \in [N]} (T(x_i) - \frac{1}{N} \sum_{j \in [N]} T(x_j))(T(x_i) - \frac{1}{N} \sum_{j \in [N]} T(x_j))^\top. \tag{110}$$

Hence,

$$\frac{\partial}{\partial \tau} \left\{ \text{var}(\mathbf{f}(\breve{\theta}(\tau))) \right\} = \left( \frac{\partial}{\partial \tau} \breve{\theta}(\tau) \right)^\top \nabla_\theta \left\{ \text{var}(\mathbf{f}(\breve{\theta}(\tau))) \right\} \tag{111}$$

$$= 2 \left( \frac{\partial}{\partial \tau} \breve{\theta}(\tau) \right)^\top \mathcal{V}_0 \breve{\theta}(\tau) \tag{112}$$

$$= -2\breve{\theta}^\top(\tau) \mathcal{V}(\tau; \breve{\theta}(\tau)) \left( \nabla_{\theta\theta}^2 A(\breve{\theta}(\tau)) + \tau \mathcal{V}(\tau; \breve{\theta}(\tau)) \right)^{-1} \mathcal{V}_0 \breve{\theta}(\tau) \tag{113}$$

$$< 0, \tag{114}$$

completing the proof. $\square$

**Theorem 5** (Cosine similarity of the loss vector and the all-ones vector increases with $t$). *For* $\mathbf{u}, \mathbf{v} \in \mathbb{R}^N$, *let cosine similarity be defined as*

$$s(\mathbf{u}, \mathbf{v}) := \frac{\mathbf{u}^\top \mathbf{v}}{\|\mathbf{u}\|_2 \|\mathbf{v}\|_2}. \tag{115}$$

*Let $\mathbf{f}(\theta) := (f(x_1; \theta)), \ldots, f(x_N; \theta))$ and let $\mathbf{1}_N$ denote the all-1 vector of length $N$. Then, under Assumption 3 and Assumption 4, for any $t \in \mathbb{R}$,*

$$\frac{\partial}{\partial t} \left\{ s(\mathbf{f}(\breve{\theta}(t)), \mathbf{1}_N) \right\} > 0. \tag{116}$$

*Proof.* Notice that

$$s(\mathbf{f}(\theta), \mathbf{1}_N) = \frac{\frac{1}{N} \sum_{i \in [N]} f(x_i; \theta)}{\sqrt{\frac{1}{N} \sum_{i \in [N]} f^2(x_i; \theta)}}. \tag{117}$$

Let $\mathcal{M}_0 := \mathcal{M}(0; \theta)$ and $\mathcal{V}_0 := \mathcal{V}(0; \theta)$. Hence,

$$\frac{1}{N} \sum_{i \in [N]} f(x_i; \theta) = A(\theta) - \theta^\top \mathcal{M}_0, \tag{118}$$

$$\frac{1}{N} \sum_{i \in [N]} f^2(x_i; \theta) = (A(\theta) - \theta^\top \mathcal{M}_0)^2 + \theta^\top \mathcal{V}_0 \theta \tag{119}$$

Notice that

$$\nabla_\theta \left\{ s^2(\mathbf{f}(\theta), \mathbf{1}_N) \right\} = \nabla_\theta \left\{ \frac{\left( \frac{1}{N} \sum_{i \in [N]} f(x_i; \theta) \right)^2}{\frac{1}{N} \sum_{i \in [N]} f^2(x_i; \theta)} \right\} \tag{120}$$

$$= \nabla_\theta \left\{ \frac{(A(\theta) - \theta^\top \mathcal{M}_0)^2}{(A(\theta) - \theta^\top \mathcal{M}_0)^2 + \theta^\top \mathcal{V}_0 \theta} \right\} \tag{121}$$

$$= \frac{2(A(\theta) - \theta^\top \mathcal{M}_0)(\nabla_\theta A(\theta) - \mathcal{M}_0)\theta^\top \mathcal{V}_0 \theta - 2(A(\theta) - \theta^\top \mathcal{M}_0)^2 \mathcal{V}_0 \theta}{\left( (A(\theta) - \theta^\top \mathcal{M}_0)^2 + \theta^\top \mathcal{V}_0 \theta \right)^2} \tag{122}$$

$$= \frac{2(A(\theta) - \theta^\top \mathcal{M}_0) \left( \theta^\top (\nabla_\theta A(\theta) - \mathcal{M}_0) - A(\theta) + \theta^\top \mathcal{M}_0 \right) \mathcal{V}_0 \theta}{\left( (A(\theta) - \theta^\top \mathcal{M}_0)^2 + \theta^\top \mathcal{V}_0 \theta \right)^2} \tag{123}$$

$$= \frac{2(A(\theta) - \theta^\top \mathcal{M}_0) \left( \theta^\top \nabla_\theta A(\theta) - A(\theta) \right) \mathcal{V}_0 \theta}{\left( (A(\theta) - \theta^\top \mathcal{M}_0)^2 + \theta^\top \mathcal{V}_0 \theta \right)^2} \tag{124}$$

$$= -\frac{2(A(\theta) - \theta^\top \mathcal{M}_0)^2 \mathcal{V}_0 \theta}{\left( (A(\theta) - \theta^\top \mathcal{M}_0)^2 + \theta^\top \mathcal{V}_0 \theta \right)^2}. \tag{125}$$

Hence,

$$\frac{\partial}{\partial \tau} \left\{ s^2(\mathbf{f}(\breve{\theta}(\tau)), \mathbf{1}_N) \right\} = \left( \frac{\partial}{\partial \tau} \breve{\theta}(\tau) \right)^\top \nabla_\theta \left\{ s^2(\mathbf{f}(\breve{\theta}(\tau)), \mathbf{1}_N) \right\} \tag{126}$$

$$= -\breve{\theta}^\top(\tau) \mathcal{V}(\tau; \breve{\theta}(\tau)) \left( \nabla_{\theta\theta}^2 A(\breve{\theta}(\tau)) + \tau \mathcal{V}(\tau; \breve{\theta}(\tau)) \right)^{-1}$$

$$\times - \frac{2(A(\breve{\theta}(\tau)) - \breve{\theta}(\tau)^\top \mathcal{M}_0)^2}{\left( (A(\breve{\theta}(\tau)) - \breve{\theta}(\tau)^\top \mathcal{M}_0)^2 + \breve{\theta}(\tau)^\top \mathcal{V}_0 \theta \right)^2} \mathcal{V}_0 \breve{\theta}(\tau) \tag{127}$$

$$> 0, \tag{128}$$

completing the proof. $\qquad \square$

**Theorem 6** (Gradient weights become more uniform by increasing $t$). *Under Assumption 3 and Assumption 4, for any $\tau, t \in \mathbb{R}$,*

$$\frac{\partial}{\partial t} H(\mathbf{w}(\tau; \breve{\theta}(t))) > 0, \tag{129}$$

*where $H(\cdot)$ denotes the Shannon entropy function measured in nats,*

$$H(\mathbf{w}(t; \theta)) := -\sum_{i \in [N]} w_i(t; \theta) \log w_i(t; \theta). \tag{130}$$

*Proof.* Notice that

$$H\left(\mathbf{w}(t;\theta)\right) = -\sum_{i\in[N]} w_i(t;\theta)\log w_i(t;\theta) \tag{131}$$

$$= -\sum_{i\in[N]} (tf(x_i;\theta) - \Lambda(t;\theta))e^{tf(x_i;\theta)-\Lambda(t;\theta)} \tag{132}$$

$$= \Lambda(t;\theta) - t\sum_{i\in[N]} f(x_i;\theta)e^{tf(x_i;\theta)-\Lambda(t;\theta)} \tag{133}$$

$$= \Lambda(t;\theta) - tA(\theta) + t\theta^\top \mathcal{M}(t;\theta). \tag{134}$$

Thus,

$$\nabla_\theta H\left(\mathbf{w}(t;\theta)\right) = \nabla_\theta \left(\Lambda(t;\theta) - tA(\theta) + t\theta^\top \mathcal{M}(t;\theta)\right) \tag{135}$$

$$= t\nabla_\theta A(\theta) - t\mathcal{M}(t;\theta) - t\nabla_\theta A(\theta) + t\mathcal{M}(t;\theta) - t^2\mathcal{V}(t;\theta)\theta \tag{136}$$

$$= -t^2\mathcal{V}(t;\theta)\theta. \tag{137}$$

Hence,

$$\frac{\partial}{\partial\tau}H\left(\mathbf{w}(t;\breve{\theta}(\tau))\right) = \left(\frac{\partial}{\partial\tau}\breve{\theta}(\tau)\right)^\top \nabla_\theta H\left(\mathbf{w}(t;\breve{\theta}(\tau))\right) \tag{138}$$

$$= \nabla_\theta \left(\Lambda(t;\theta) - tA(\theta) + t\theta^\top \mathcal{M}(t;\theta)\right) \tag{139}$$

$$= t^2\breve{\theta}^\top(\tau)\mathcal{V}(\tau;\breve{\theta}(\tau))\left(\nabla^2_{\theta\theta}A(\breve{\theta}(\tau)) + \tau\mathcal{V}(\tau;\breve{\theta}(\tau))\right)^{-1}\mathcal{V}(t;\breve{\theta}(\tau))\breve{\theta}(\tau) \tag{140}$$

$$\geqslant 0, \tag{141}$$

completing the proof. $\qquad\square$

**Theorem 7** (Tilted objective is increasing with $t$). *Under Assumption 3, for all $t \in \mathbb{R}$, and all $\theta \in \Theta$,*

$$\frac{\partial}{\partial t}\widetilde{R}(t;\theta) \geqslant 0. \tag{142}$$

*Proof.* Following (56),

$$\frac{\partial}{\partial t}\widetilde{R}(t;\theta) = \frac{\partial}{\partial t}\left\{\frac{1}{t}\Gamma(t;\theta)\right\} \tag{143}$$

$$= -\frac{1}{t^2}\Gamma(t;\theta) - \frac{1}{t}\theta^\top \mathcal{M}(t;\theta), \tag{144}$$

$$=: g(t;\theta), \tag{145}$$

where (144) follows from Lemma 9, and (145) defines $g(t;\theta)$.

Let $g(0;\theta) := \lim_{t\to0} g(t;\theta)$ Notice that

$$g(0;\theta) = \lim_{t\to0}\left\{-\frac{1}{t^2}\Gamma(t;\theta) - \frac{1}{t}\theta^\top \mathcal{M}(t;\theta)\right\} \tag{146}$$

$$= -\lim_{t\to0}\left\{\frac{\frac{1}{t}\Gamma(t;\theta) + \theta^\top \mathcal{M}(t;\theta)}{t}\right\} \tag{147}$$

$$= \theta^\top \mathcal{V}(0;\theta)\theta, \tag{148}$$

where (148) is due to L'Hôpital's rule and Lemma 12. Now consider

$$\frac{\partial}{\partial t}\left\{t^2 g(t;\theta)\right\} = \frac{\partial}{\partial t}\left\{-\Gamma(t;\theta) - t\theta^\top \mathcal{M}(t;\theta)\right\} \tag{149}$$

$$= \theta^\top \mathcal{M}(t;\theta) \tag{150}$$

$$\quad - \theta^\top \mathcal{M}(t;\theta) + t\theta^\top \mathcal{V}(t;\theta)\theta \tag{151}$$

$$= t\theta^\top \mathcal{V}(t;\theta)\theta \tag{152}$$

where $g(t;\theta) = \frac{\partial}{\partial t}\widetilde{R}(t;\theta)$, (150) follows from Lemma 9, (151) follows from the chain rule and Lemma 10. Hence, $t^2 g(t;\theta)$ is an increasing function of $t$ for $t \in \mathbb{R}^+$, and a decreasing function of $t$ for $t \in \mathbb{R}^-$, taking its minimum at $t = 0$. Hence, $t^2 g(t;\theta) \geqslant 0$ for all $t \in \mathbb{R}$. This implies that $g(t;\theta) \geqslant 0$ for all $t \in \mathbb{R}$, which in conjunction with (145) implies the statement of the theorem. $\qquad\square$

**Definition 4** (Optimal tilted objective). *Let the optimal tilted objective be defined as*

$$\widetilde{F}(t) := \widetilde{R}(t;\breve{\theta}(t)). \tag{153}$$

**Theorem 8** (Optimal tilted objective is increasing with $t$). *Under Assumption 3, for all $t \in \mathbb{R}$, and all $\theta \in \Theta$,*

$$\frac{\partial}{\partial t}\widetilde{F}(t) = \frac{\partial}{\partial t}\widetilde{R}(t;\breve{\theta}(t)) \geqslant 0. \tag{154}$$

*Proof.* Notice that for all $\theta$, and all $\epsilon \in \mathbb{R}^+$,

$$\widetilde{R}(t + \epsilon;\theta) \geqslant \widetilde{R}(t;\theta) \tag{155}$$
$$\geqslant \widetilde{R}(t;\breve{\theta}(t)), \tag{156}$$

where (155) follows from Theorem 7 and (156) follows from the definition of $\breve{\theta}(t)$. Hence,

$$\widetilde{R}(t + \epsilon;\breve{\theta}(t + \epsilon)) = \min_{\theta \in B(\breve{\theta}(t),r)} \widetilde{R}(t + \epsilon;\theta) \geqslant \widetilde{R}(t;\breve{\theta}(t)), \tag{157}$$

which completes the proof. $\qquad\square$

# F  CONNECTIONS BETWEEN TERM AND EXPONENTIAL TILTING

Here we provide connections between TERM and exponential tilting, a concept previously explored in the context of importance sampling and the theory of large deviations (Beirami et al., 2018; Dembo & Zeitouni, 2009; Wainwright et al., 2005). To do so, suppose that $X$ is drawn from distribution $p(\cdot)$. Let us study the distribution of random variable $Y = f(X; \theta)$. Let $\Lambda_Y(t)$ be the cumulant generating function (Dembo & Zeitouni, 2009, Sectiom 2.2). That is

$$\Lambda_Y(t) := \log\left(E_p\left\{e^{tY}\right\}\right) \tag{158}$$

$$= \log\left(E_p\left\{e^{tf(X;\theta)}\right\}\right). \tag{159}$$

Now, suppose that $x_1, \ldots, x_N$ are drawn i.i.d. from $p(\cdot)$. *Note that this distributional assumption is made solely for providing intuition on the tilted objectives and is not needed in any of the proofs in this paper*. Hence, $\widetilde{R}(t; \theta)$ can be viewed as an empirical approximation to the cumulant generating function:

$$\Lambda_Y(t) \approx t\widetilde{R}(t; \theta). \tag{160}$$

Hence, $\widetilde{R}(t; \theta)$ provides an approximate characterization of the distribution of $f(X; \theta)$. Thus, minimizing $\widetilde{R}(t; \theta)$ is approximately equivalent to minimizing the complementary cumulative distribution function (CDF) of $f(X; \theta)$. In other words, this is equivalent to minimizing $P\{f(X; \theta) > a\}$ for some $a$, which is a function of $t$.

In the next section, we will explore these connections with tail probabilities dropping the distributional assumptions, effectively drawing connections between superquantile methods and TERM.

## G  TERM AS AN APPROXIMATE SUPERQUANTILE METHOD

For all $a \in \mathbb{R}$, let $Q(a; \theta)$ denote the quantile of the losses that are no smaller than $a$, i.e.,

$$Q(a; \theta) := \frac{1}{N} \sum_{i \in [N]} \mathbb{I} \{f(x_i; \theta) \geqslant a\}, \tag{161}$$

where $\mathbb{I}\{\cdot\}$ is the indicator function. Notice that $Q(a; \theta) \in \left\{0, \frac{1}{N}, \dots, 1\right\}$ quantifies the fraction of the data for which loss is at least $a$. In this section, we further assume that $f$ is such that $f(x_i; \theta) \geqslant 0$ for all $\theta$.

Suppose that we are interested in choosing $\theta$ in a way that for a given $a \in \mathbb{R}$, we minimize the fraction of the losses that are larger than $a$. That is

$$Q^0(a) := \min_\theta Q(a; \theta) = Q(a; \theta^0(a)), \tag{162}$$

where

$$\theta^0(a) := \arg\min_\theta Q(a; \theta). \tag{163}$$

This is a non-smooth non-convex problem and solving it to global optimality is very challenging. In this section, we argue that TERM provides a reasonable approximate solution to this problem, which is computationally feasible.

Notice that we have the following simple relation:

**Lemma 20.** *If $a < \widetilde{F}(-\infty)$ then $Q^0(a) = 1$. Further, if $a > \widetilde{F}(+\infty)$ then $Q^0(a) = 0$, where $\widetilde{F}(\cdot)$ is defined in Definition 4, and is reproduced here:*

$$\widetilde{F}(-\infty) = \lim_{t \to -\infty} \widetilde{R}(t; \breve{\theta}(t)) = \min_\theta \min_{i \in [N]} f(x_i; \theta), \tag{164}$$

$$\widetilde{F}(+\infty) = \lim_{t \to +\infty} \widetilde{R}(t; \breve{\theta}(t)) = \min_\theta \max_{i \in [N]} f(x_i; \theta). \tag{165}$$

Next, we present our main result on the connection between the superquantile method and TERM.

**Theorem 9.** *For all $t \in \mathbb{R}$, and all $\theta$, and all $a \in (\widetilde{F}(-\infty), \widetilde{F}(+\infty))$,[3]*

$$Q(a; \theta) \leqslant \widetilde{Q}(a; t, \theta) := \frac{e^{\widetilde{R}(t; \theta)t} - e^{\widetilde{F}(-\infty)t}}{e^{at} - e^{\widetilde{F}(-\infty)t}}. \tag{166}$$

*Proof.* We have

$$Q(a; \theta) = \frac{\frac{1}{N} \sum_{i \in [N]} e^{(a - \widetilde{F}(-\infty))t \mathbb{I}\left\{\frac{f(x_i; \theta) - \widetilde{F}(-\infty)}{a - \widetilde{F}(-\infty)} \geqslant 1\right\}} - 1}{e^{(a - \widetilde{F}(-\infty))t} - 1} \tag{167}$$

$$\leqslant \frac{\frac{1}{N} \sum_{i \in [N]} e^{(f(x_i; \theta) - \widetilde{F}(-\infty))t} - 1}{e^{(a - \widetilde{F}(-\infty))t} - 1} \tag{168}$$

$$= \frac{e^{\widetilde{R}(t; \theta)t} - e^{\widetilde{F}(-\infty)t}}{e^{at} - e^{\widetilde{F}(-\infty)t}}, \tag{169}$$

where (167) follows from Lemma 21, (168) follows from Lemma 22, the fact that $e^{tx}$ is strictly increasing (resp. decreasing) for $t > 0$ (resp. $t < 0$) and $(e^{(a - \widetilde{F}(-\infty))t} - 1)$ is positive (resp. negative) for $t > 0$ (resp. $t < 0$), and (169) follows from definition. $\qquad\square$

**Lemma 21.** *For all $t \in \mathbb{R}$, and all $\theta$,[4]*

$$Q(a; \theta) = \frac{\frac{1}{N} \sum_{i \in [N]} e^{(a - \widetilde{F}(-\infty))t \mathbb{I}\{f(x_i; \theta) \geqslant a\}} - 1}{e^{(a - \widetilde{F}(-\infty))t} - 1}. \tag{170}$$

---

[3]We define the RHS at $t = 0$ via continuous extension.

[4]We define the RHS at $t = 0$ via continuous extension.

*Proof.* The proof is completed following this identity:

$$\frac{1}{N} \sum_{i \in [N]} e^{(a - \widetilde{F}(-\infty))\mathbb{I}\{f(x_i; \theta) \geqslant a\}t} = Q(a; \theta) e^{(a - \widetilde{F}(-\infty))t} + (1 - Q(a; \theta)). \tag{171}$$

$\square$

**Lemma 22.** *For $x \geqslant 0$, we have $\mathbb{I}\{x \geqslant 1\} \leqslant x$.*

Theorem 9 directly leads to the following result.

**Theorem 10.** *For all $a \in (\widetilde{F}(-\infty), \widetilde{F}(+\infty))$, we have*

$$Q^0(a) \leqslant Q^1(a) \leqslant Q^2(a) \leqslant Q^3(a) = \inf_{t \in \mathbb{R}} \left\{ \frac{e^{\widetilde{F}(t)t} - e^{\widetilde{F}(-\infty)t}}{e^{at} - e^{\widetilde{F}(-\infty)t}} \right\}, \tag{172}$$

*where*

$$Q^1(a) := \inf_{t \in \mathbb{R}} Q(a; \breve{\theta}(t)) \tag{173}$$

$$Q^2(a) := Q(a; \breve{\theta}(\widetilde{t}(a))) \tag{174}$$

$$Q^3(a) := \widetilde{Q}(a; \widetilde{t}(a), \breve{\theta}(\widetilde{t}(a))) \tag{175}$$

*and*

$$\widetilde{t}(a) := \arg\inf_{t \in \mathbb{R}} \left\{ \widetilde{Q}(a; t, \breve{\theta}(t)) \right\} = \arg\inf_{t \in \mathbb{R}} \left\{ \frac{e^{\widetilde{F}(t)t} - e^{\widetilde{F}(-\infty)t}}{e^{at} - e^{\widetilde{F}(-\infty)t}} \right\}. \tag{176}$$

*Proof.* The only non-trivial step is to show that $Q^2(a) \leqslant Q^3(a)$. Following Theorem 9,

$$Q^2(a) = Q(a; \breve{\theta}(\widetilde{t}(a))) \tag{177}$$

$$\leqslant \inf_{t \in \mathbb{R}} \widetilde{Q}(a; t, \breve{\theta}(t)) \tag{178}$$

$$= Q^3(a), \tag{179}$$

which completes the proof. $\square$

Theorem 10 motivates us with the following approximation on the solutions of the superquantile method.

**Approximation 1.** *For all $a \in (\widetilde{F}(-\infty), \widetilde{F}(+\infty))$,*

$$Q(a; \theta^0(a)) = Q^0(a) \approx Q^2(a) = Q(a; \breve{\theta}(\widetilde{t}(a))), \tag{180}$$

*and hence, $\breve{\theta}(\widetilde{t}(a)$ is an approximate solution to the superquantile optimization problem.*

While we have not characterized how tight this approximation is for $a \in (\widetilde{F}(-\infty), \widetilde{F}(+\infty))$, we believe that Approximation 1 provides a reasonable solution to the superquantile optimization problem in general. This is evidenced empirically when the approximation is evaluated on the toy examples of Figure 1, and compared with the global solutions of the superquantile method. The results are shown in Figure 6. As can be seen, $Q^0(a) \approx Q^2(a)$ as suggested by Approximation 1. Also, we can see that while the bound in Theorem 10 is not tight, the solution that is obtained from solving it results in a good approximation to the superquantile optimization.

Finally, we draw connections between these results and the $k$-loss. Notice that minimizing $Q(a; \theta)$ for a fixed $a$ is equivalent to minimizing $a$ for a fixed $Q(a; \theta)$. If we fix $Q(a; \theta) = (N - k)/N$, minimizing $a$ would be equivalent to minimizing the $k$-loss. Formally, let $R_{(k)}(\theta)$ be the $k$-th order statistc of the loss vector. Hence, $R_{(k)}$ is the $k$-th smallest loss, and particularly

$$R_{(1)}(\theta) = \breve{R}(\theta), \tag{181}$$

$$R_{(N)}(\theta) = \widehat{R}(\theta). \tag{182}$$

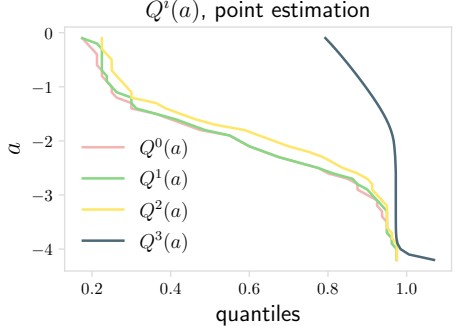
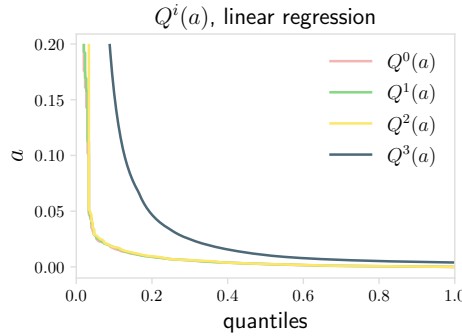

(a) Numerical results showing the bounds $Q^1(a), Q^2(a)$, and $Q^3(a)$ for $Q^0(a)$ on the point estimation example.

(b) Numerical results showing the bounds $Q^1(a), Q^2(a)$, and $Q^3(a)$ for $Q^0(a)$ on the linear regression example.

Figure 6: $Q^1(a)$ and $Q^2(a)$ are close to $Q^0(a)$, which indicates that the solution obtained from solving $Q^3(a)$ (which is $Q^2(a)$) is a tight approximation of the globally optimal solution of $Q^0(a)$.

Thus, for any $k \in [N]$, we define

$$R^*_{(k)} := \min_\theta R_{(k)}(\theta). \tag{183}$$

$$\theta^*(k) := \arg\min_\theta R_{(k)}(\theta). \tag{184}$$

Note that

$$R^*_{(1)} = \widetilde{F}(-\infty), \tag{185}$$

$$R^*_{(N)} = \widetilde{F}(+\infty). \tag{186}$$

Theorem 9 directly implies the following result:

**Corollary 11.** *For all $k \in \{2, \ldots, N-1\}$, and all $t \in \mathbb{R}^+$ :*

$$\left| e^{(R_{(k)}(\theta) - \widetilde{F}(-\infty))t} - 1 \right| \leqslant \left( \frac{N}{N-k} \right) \left| e^{(\widetilde{R}(t;\theta) - \widetilde{F}(-\infty))t} - 1 \right|. \tag{187}$$

*Proof.* We proceed by setting $Q(a; \theta) = \frac{N-k}{N}$ and $a = R_{(k)}(\theta)$ in Theorem 9, which implies the result. $\square$

While the bound is left implicit in Corollary 11, we can obtain an explicit bound if we only consider $t \in \mathbb{R}^+$ (i.e., we are interested in $k$-losses for larger $k$):

**Corollary 12.** *For all $k \in \{2, \ldots, N-1\}$, and all $t \in \mathbb{R}^+$ :*

$$R_{(k)}(\theta) \leqslant \widetilde{F}(-\infty) + \frac{1}{t} \log \left( \frac{e^{(\widetilde{R}(t;\theta) - \widetilde{F}(-\infty))t} - \frac{k}{N}}{1 - \frac{k}{N}} \right). \tag{188}$$

*Proof.* The statement follows by algebraic manipulation of Corollary 11. $\square$

# H  ALGORITHMS FOR SOLVING TERM

In the main text, we present TERM in the batch setting (Algorithm 1). Here we provide the stochastic variants of the solvers in the context of hierarchical multi-objective tilting (see Eq. (4)).

There are a few points to note about stochastic solvers (Algorithm 2):

1. It is intractable to compute the exact normalization weights for the samples in the minibatch. Hence, we use $\widetilde{R}_{g,\tau}$, a term that incorporates stochastic dynamics, to follow the tilted objective for each group $g$, which is used for normalizing the weights as in (3).

2. While we sample the group from which we draw the minibatch, for small number of groups, one might want to draw one minibatch per each group and weight the resulting gradients accordingly.

3. The second last line in Algorithm 2, concerning the update of $\widetilde{R}_{g,\tau}$ is not a trivial linear averaging. Instead, we use a tilted averaging to ensure an unbiased estimator (if $\theta$ is not being updated).

---

**Algorithm 2:** Stochastic TERM

---

**Initialize :** $\widetilde{R}_{g,\tau} = 0 \; \forall g \in [G]$
**Input:** $t, \tau, \alpha, \lambda$
**while** *stopping criteria not reached* **do**
     sample $g$ on $[G]$ from a Gumbel-Softmax distribution with logits $\widetilde{R}_{g,\tau} + \frac{1}{t} \log |g|$ and
       temperature $\frac{1}{t}$
     sample minibatch $B$ uniformly at random within group $g$
     compute the loss $f(x; \theta)$ and gradient $\nabla_\theta f(x; \theta)$ for all $x \in B$
     $\widetilde{R}_{B,\tau} \leftarrow \tau$-tilted loss (2) on minibatch $B$
     $\widetilde{R}_{g,\tau} \leftarrow \frac{1}{\tau} \log \left( (1-\lambda)e^{\tau \widetilde{R}_{g,\tau}} + \lambda e^{\tau \widetilde{R}_{B,\tau}} \right), \; w_{\tau,x} \leftarrow e^{\tau f(x;\theta) - \tau \widetilde{R}_{g,\tau}}$
     $\theta \leftarrow \theta - \frac{\alpha}{|B|} \sum_{x \in B} w_{\tau,x} \nabla_\theta f(x; \theta)$
**end**

---

The stochatic algorithm above requires roughly the same time/space complexity as mini-batch SGD, and thus scales similarly for large-scale problems. TERM for the non-hierarchical cases can be recovered from Algorithm 1 and 2 by setting the inner-level tilt parameter $\tau = 0$. For completeness, we also describe them here. Algorithm 3 is the sample-level tilting algorithm in the batch setting, and Algorithm 4 is its stochastic variant.

---

**Algorithm 3:** Batch Non-Hierarchical TERM

---

**Input:** $t, \alpha$
**while** *stopping criteria not reached* **do**
     compute the loss $f(x_i; \theta)$ and gradient $\nabla_\theta f(x_i; \theta)$ for all $i \in [N]$
     $\widetilde{R}(t; \theta) \leftarrow t$-tilted loss (2) on all $i \in [N]$
     $w_i(t; \theta) \leftarrow e^{t(f(x_i;\theta) - \widetilde{R}(t;\theta))}$
     $\theta \leftarrow \theta - \frac{\alpha}{N} \sum_{i \in [N]} w_i(t; \theta) \nabla_\theta f(x_i; \theta)$
**end**

---

In order to verify the correctness of Algorithm 4, we plot the distance of the solution $\theta$ to the solution $\theta^*$ obtained by running the full gradient method (Algorithm 1) in terms of the number of iterations. In Figure 7, we see that $\theta$ can be arbitrarily close to $\theta^*$, and Algorithm 4 with $t \neq 0$ converges similarly to mini-batch SGD with $t = 0$. As mentioned in the main text, theoretically analyze the convergence of stochastic solvers would be interesting direction of future work. The challenges would be to characterize the tightness of the estimator $\widetilde{R}$ to the true risk $R$ at each iteration leveraging the proposed tilted averaging structure.

We summarize the applications we solve TERM for and the algorithms we use in Table 4 below on the left.

---

**Algorithm 4:** Stochastic Non-Hierarchical TERM

---

**Initialize :** $\widetilde{R}_t = 0$
**Input:** $t, \alpha, \lambda$
**while** *stopping criteria not reached* **do**
    sample minibatch $B$ uniformly at random from $[N]$
    compute the loss $f(x; \theta)$ and gradient $\nabla_\theta f(x; \theta)$ for all $x \in B$
    $\widetilde{R}_{B,t} \leftarrow t$-tilted loss (2) on minibatch $B$
    $\widetilde{R}_t \leftarrow \frac{1}{t} \log \left( (1 - \lambda) e^{t\widetilde{R}_t} + \lambda e^{t\widetilde{R}_{B,t}} \right), w_{t,x} \leftarrow e^{tf(x;\theta) - t\widetilde{R}_t}$
    $\theta \leftarrow \theta - \frac{\alpha}{|B|} \sum_{x \in B} w_{t,x} \nabla_\theta f(x; \theta)$
**end**

---

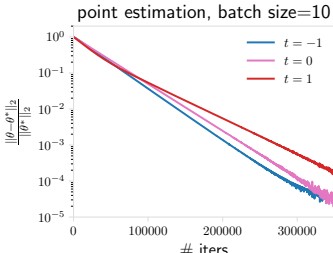

point estimation, batch size=10

| Three toy examples (Figure 1) | Algorithm 3 |
|---|---|
| Robust Regression (Table 1) | Algorithm 3 |
| Robust Classification (Table 2) | Algorithm 4 |
| Low-quality Annotators (Figure 3) | Algorithm 2 ($\tau = 0$) |
| Fair PCA (Figure 4) | Algorithm 1 ($\tau = 0$) |
| Class Imbalance (Figure 5) | Algorithm 2 ($\tau = 0$) |
| Variance Reduction (Table 8) | Algorithm 1 ($\tau = 0$) |
| Hierarchical TERM (Table 3) | Algorithm 1 |

Figure 7: Correctness of Algorithm 4. The y-axis is the normalized distance between $\theta$ obtained by running Alg. 4 and the optimal solution $\theta^*$ via the full batch method. For different values of $t$, $\theta$ can be arbitrarily close to $\theta^*$.

Table 4: Applications and their corresponding solvers.

## H.1 CONVERGENCE WITH $t$

First, we note that $t$-tilted losses are $\beta(t)$-smooth for all $t$. In a small neighborhood around the tilted solution, $\beta(t)$ is bounded for all negative $t$ and moderately positive $t$, whereas it scales linearly with $t$ as $t \to +\infty$, which has been previously studied in the context of exponential smoothing of the max (Kort & Bertsekas, 1972; Pee & Royset, 2011). We prove this formally in Appendix B, Lemma 4, but it can also be observed visually via the toy example in Figure 2. Based on this, we provide a convergence result below for Algorithm 3.

**Theorem 13.** *Under Assumption 2, there exist $C_1, C_2 < \infty$ that do not depend on $t$ such that for any $t \in \mathbb{R}^+$, setting the step size $\alpha = \frac{1}{C_1 + C_2 t}$, after $m$ iterations:*

$$\widetilde{R}(t, \theta_m) - \widetilde{R}(t, \breve{\theta}(t)) \leqslant \left( 1 - \frac{\beta_{\min}}{C_1 + C_2 t} \right)^m \left( \widetilde{R}(t, \theta_0) - \widetilde{R}(t, \breve{\theta}(t)) \right). \quad (189)$$

*Proof.* First observe that by Lemma 5, $\widetilde{R}(t, \theta)$ is $\beta_{\min}$-strongly convex for all $t \in \mathbb{R}^+$. Next, notice that by Lemma 4, there exist $C_1, C_2 < \infty$ such that $\widetilde{R}(t; \theta)$ has a $(C_1 + C_2 t)$-Lipschitz gradient for all $t \in \mathbb{R}^+$. Then, the result follows directly from (Karimi et al., 2016)[Theorem 1]. $\square$

Theorem 13 indicates that solving TERM to a local optimum using gradient-based methods will tend to be as efficient as traditional ERM for small-to-moderate values of $t$ (Jin et al., 2017), which we corroborate via experiments on multiple real-world datasets in Section 5. This is in contrast to solving for the min-max solution, which would be similar to solving TERM as $t \to +\infty$ (Kort & Bertsekas, 1972; Ostrovskii et al., 2020; Pee & Royset, 2011).

Second, recall that the $t$-tilted loss remains strongly convex for $t > 0$, so long as the origi-

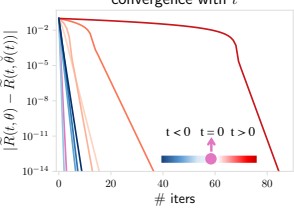

convergence with $t$

Figure 8: As $t \to +\infty$, the objective becomes less smooth in the vicinity of the final solution, hence suffering from slower convergence. For negative values of $t$, TERM converges fast due to the smoothness in the vicinity of solutions despite its non-convexity.

nal loss function is strongly convex. On the other hand, for sufficiently large negative $t$, the $t$-tilted loss becomes non-convex. Hence, while the $t$-tilted solutions for positive $t$ are unique, the objective may have multiple (spurious) local minima for negative $t$ even if the original loss function is strongly convex. For negative $t$, we seek the solution for which the parametric set of $t$-tilted solutions obtained by sweeping $t \in \mathbb{R}$ remains continuous (as in Figure 1a-c). To this end, for negative $t$, we solve TERM by smoothly decreasing $t$ from $0$ ensuring that the solutions form a continuum in $\mathbb{R}^d$. Despite the non-convexity of TERM with $t < 0$, we find that this approach produces effective solutions to multiple real-world problems in Section 5. Additionally, as the objective remains smooth, it is still relatively efficient to solve. We plot the convergence with $t$ on a toy problem in Figure 8.

# I    ADDITIONAL EXPERIMENTS

In this section we provide complete experimental results showcasing the properties of TERM (Appendix I.1) and the use-cases covered in Section 5 (Appendix I.2). Details on how the experiments themselves were executed are provided in Appendix J.

## I.1    EXPERIMENTS TO SHOWCASE PROPERTIES OF TERM

Recall that in Section 2, Interpretation 1 is that TERM can be tuned to re-weight samples to magnify or suppress the influence of outliers. In Figure 9 below, we visually show this effect by highlighting the samples with the largest weight for $t \to +\infty$ and $t \to -\infty$ on the logistic regression example previously described in Figure 1.

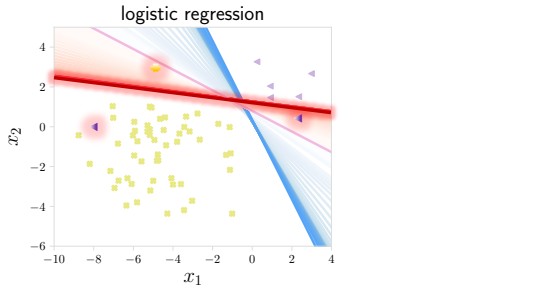
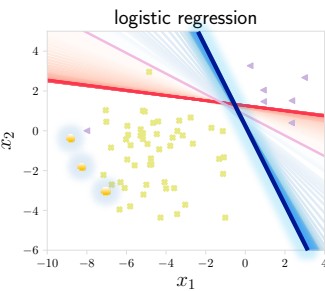

(a) Samples with the largest weights when $t \to +\infty$.    (b) Samples with the largest weights when $t \to -\infty$.

Figure 9: For positive values of $t$, TERM focuses on the samples with relatively large losses (rare instances). When $t \to +\infty$ (left), a few misclassified samples have the largest weights and are highlighted. On the other hand, for negative values of $t$, TERM suppresses the effect of the outliers, and as $t \to -\infty$ (right), samples with the smallest losses hold the the largest weights.

Interpretation 2 is concerned with smooth tradeoffs between the average-loss and max/min-loss. In Figure 10 below, we show that (1) tilted solutions with positive $t$'s achieve a smooth tradeoff between average-loss and max-loss, (2) similarly, negative $t$'s result in a smooth tradeoff between average-loss and min-loss, and (3) increasing $t$ from $-\infty$ to $+\infty$ reduces the variance of the losses.

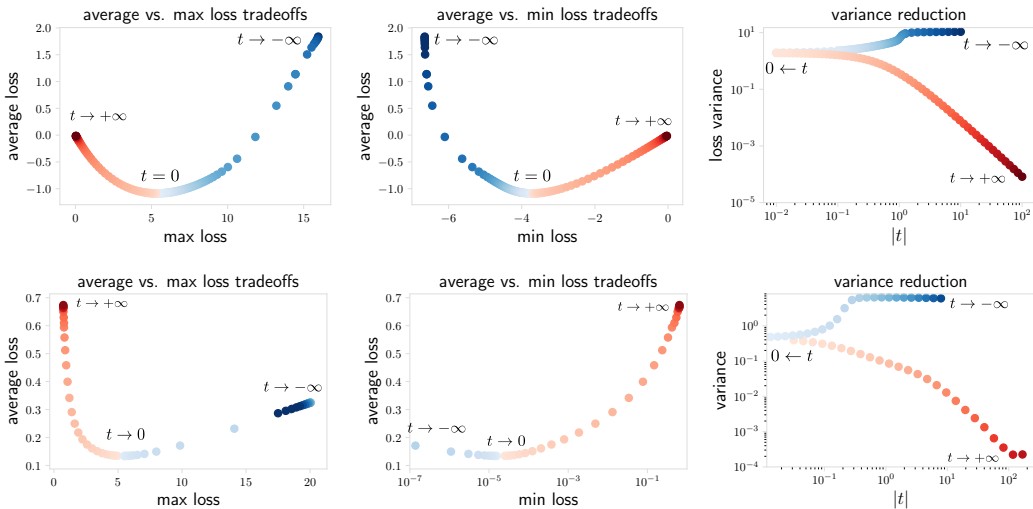

Figure 10: The tradeoffs between the average-loss and the max/min-loss offered by TERM on the point estimation (top) and logistic regression (bottom) toy examples presented in Figure 1, empirically validating Theorems 1– 4. Positive values of $t$ trade the average-loss for the max-loss, while negative values of $t$ trade the average-loss for the min-loss. Increasing $t$ from $-\infty$ to $+\infty$ results in the reduction of loss variance, allowing the solution to tradeoff between bias/variance and potentially improve generalization.

### I.2    COMPLETE CASE STUDIES

Here we provide complete results obtained from applying TERM to a diverse set of applications. We either present full metrics of the empirical results discussed in Section 5, or provide additional experiments demonstrating the effects of TERM in new settings.

**Robust regression.**    In Section 5.1, we focused on noise scenarios with random label noise. Here, we present results involving both feature noise and target noise. We investigate the performance of TERM on two datasets (cal-housing (Pace & Barry, 1997) and abalone (Dua & Graff, 2019)) used in (Yu et al., 2012). Both datasets have features with 8 dimensions. We generate noisy samples following the setup in (Yu et al., 2012)—sampling 100 training samples, and randomly corrupting 5% of them by multiplying their features by 100 and multiply their targets by 10,000. From Table 5 below, we see that TERM significantly outperforms the baseline objectives in the noisy regime on both datasets.

Table 5: An alternative noise setup involving both feature noise and label noise. Similarly, TERM with $t < 0$ significantly outperforms several baseline objectives for noisy outlier mitigation.

| objectives | test RMSE (cal-housing) | | test RMSE (abalone) | |
|---|---|---|---|---|
| | clean | noisy | clean | noisy |
| ERM | 0.766 (0.023) | 239 (9) | 2.444 (0.105) | 1013 (72) |
| $L_1$ | 0.759 (0.019) | 139 (11) | 2.435 (0.021) | 1008 (117) |
| Huber (Huber, 1964) | 0.762 (0.009) | 163 (7) | 2.449 (0.018) | 922 (45) |
| CRR (Bhatia et al., 2017) | 0.766 (0.024) | 245 (8) | 2.444 (0.021) | 986 (146) |
| TERM | 0.745 (0.007) | **0.753** (0.016) | 2.477 (0.041) | **2.449** (0.028) |
| Genie ERM | 0.766 (0.023) | 0.766 (0.028) | 2.444 (0.105) | 2.450 (0.109) |

We also provide results on synthetic data across different noise levels in two settings. In Figure 11, the mean of the noise is different from the mean of the clean data, and in Figure 12, the mean of two groups of data are the same. Similarly, TERM ($t = -2$) can effectively remove outliers in the presence of random noise.

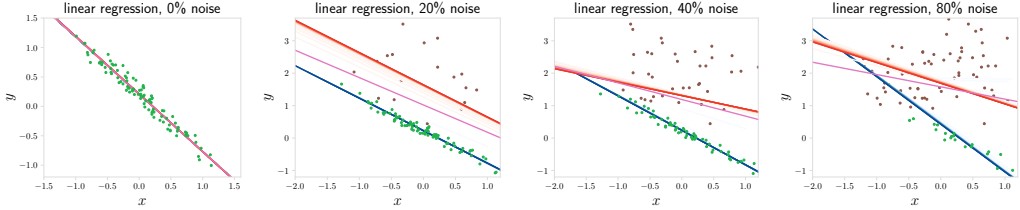

Figure 11: Robust regression on synthetic data. In the presence of random noise, TERM with negative $t$'s (blue, $t = -2$) can fit structured clean data at all noise levels, while ERM (purple) and TERM with positive $t$'s (red) overfit to corrupted data. We color inliers in green and outliers in brown for visualization purposes.

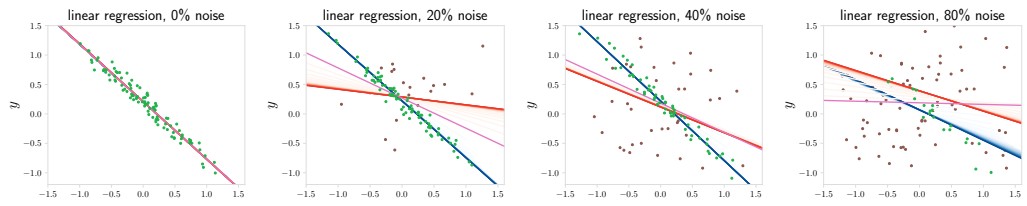

Figure 12: In the presence of random noise with the same mean as that of clean data, TERM with negative $t$'s (blue) can still surpass outliers in all cases, while ERM (purple) and TERM with positive $t$'s (red) overfit to corrupted data. While the performance drops for 80% noise, TERM can still learn useful information, and achieves much lower error than ERM.

**Robust classification.** Recall that in Section 5.1, for classification in the presence of label noise, we only compare with baselines which do not require clean validation data. In Table 6 below, we report the complete results of comparing TERM with all baselines, including MentorNet-DD (Jiang et al., 2018) which needs additional clean data. In particular, in contrast to the other methods, MentorNet-DD uses 5,000 clean validation images. TERM is competitive with can even exceed the performance of MentorNet-DD, even though it does not have access to this clean data.

Table 6: A complete comparison including two MentorNet variants. TERM is able to match the performance of MentorNet-DD, which needs additional clean labels.

| objectives | test accuracy (CIFAR-10, Inception) | | |
|---|---|---|---|
| | 20% noise | 40% noise | 80% noise |
| ERM | 0.775 (.004) | 0.719 (.004) | 0.284 (.004) |
| RandomRect (Ren et al., 2018) | 0.744 (.004) | 0.699 (.005) | 0.384 (.005) |
| SelfPaced (Kumar et al., 2010) | 0.784 (.004) | 0.733 (.004) | 0.272 (.004) |
| MentorNet-PD (Jiang et al., 2018) | 0.798 (.004) | 0.731 (.004) | 0.312 (.005) |
| GCE (Zhang & Sabuncu, 2018) | **0.805** (.004) | 0.750 (.004) | 0.433 (.005) |
| MentorNet-DD (Jiang et al., 2018) | **0.800** (.004) | **0.763** (.004) | **0.461** (.005) |
| TERM | 0.795 (.004) | **0.768** (.004) | **0.455** (.005) |
| Genie ERM | 0.828 (.004) | 0.820 (.004) | 0.792 (.004) |

To interpret the noise more easily, we provide a toy logistic regression example with synthetic data here. In Figure 13, we see that TERM with $t = -2$ (blue) can converge to the correct classifier under 20%, 40%, and 80% noise.

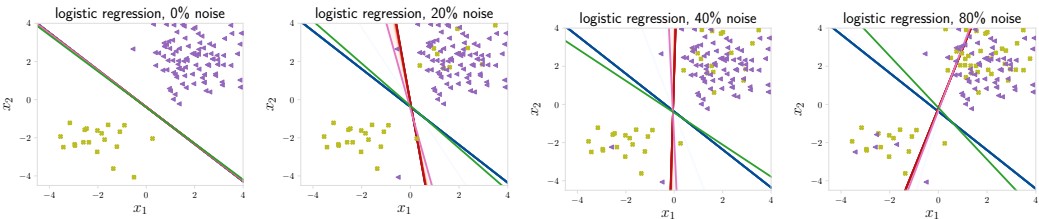

Figure 13: Robust classification using synthetic data. On this toy problem, we show that TERM with negative $t$'s (blue) can be robust to random noisy samples. The green line corresponds to the solution of the generalized cross entropy (GCE) baseline (Zhang & Sabuncu, 2018). Note that on this toy problem, GCE is as good as TERM with negative $t$'s, despite its inferior performance on the real-world CIFAR10 dataset.

(**Adversarial or structured noise**.) As a word of caution, we note that the experiments thus far have focused on random noise; as one might expect, TERM with negative $t$'s could potentially overfit to outliers if they are constructed in an adversarial way. In the examples shown in Figure 14, under 40% noise and 80% noise, TERM has a high error measured on the clean data (green dots).

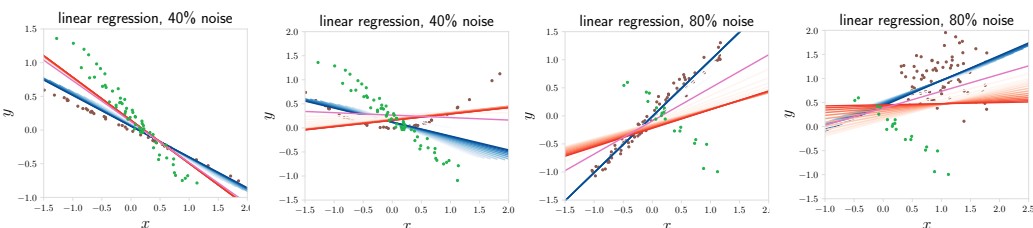

Figure 14: TERM with negative $t$'s (blue) cannot fit clean data if the noisy samples (brown) are adversarial or structured in a manner that differs substantially from the underlying true distribution.

**Low-quality annotators.** In Section 5.1, we demonstrate that TERM can be used to mitigate the effect of noisy annotators, and we assume each annotator is either always correct, or always uniformly

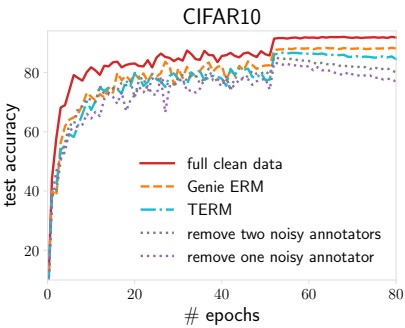

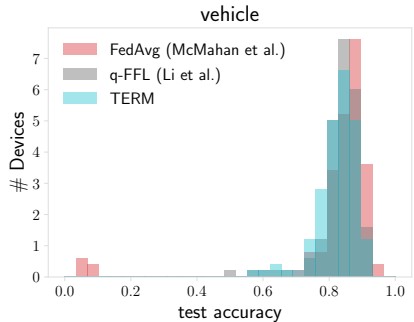

Figure 15: TERM achieves higher test accuracy than the baselines, and can match the performance of Genie ERM (i.e., training on all the clean data combined).

Figure 16: TERM FL ($t = 0.1$) significantly increases the accuracy on the worst-performing device (similar to $q$-FFL (Li et al., 2020b)) while obtaining a similar average accuracy (Table 7).

assigning random labels. Here, we explore a different and possibly more practical scenario where there are four noisy annotators who corrupt 0%, 20%, 40%, and 100% of their data by assigning labels uniformly at random, and there is one additional adversarial annotator who always assigns wrong labels. We assume the data points labeled by each annotator do not overlap, since (Khetan et al., 2018) show that obtaining one label per sample is optimal for the data collectors under a fixed annotation budget. We compare TERM with several baselines: (a) training without the data coming from the adversarial annotator, (b) training without the data coming from the worst two annotators, and (c) training with all the clean data combined (Genie ERM). The results are shown in Figure 15. We see that TERM outperforms the strong baselines of removing one or two noisy annotators, and closely matches the performance of training with all the available clean data.

**Fair federated learning.** Federated learning involves learning statistical models across massively distributed networks of remote devices or isolated organizations (Li et al., 2020a; McMahan et al., 2017). Ensuring fair (i.e., uniform) performance distribution across the devices is a major concern in federated settings (Li et al., 2020b; Mohri et al., 2019), as using current approaches for federated learning (FedAvg (McMahan et al., 2017)) may result in highly variable performance across the network. (Li et al., 2020b) consider solving an alternate objective for federated learning, called $q$-FFL, to dynamically emphasize the worst-performing devices, which is conceptually similar to the goal of TERM, though it is applied specifically to the problem of federated learning and limited to the case of positive $t$. Here, we compare TERM with $q$-FFL in their setup on the vehicle dataset (Duarte & Hu, 2004) consisting of data collected from 23 distributed sensors (hence 23 devices). We tilt the $L_2$ regularized linear SVM objective at the device level. At each communication round, we re-weight the accumulated local model updates from each selected device based on the weights estimated via Algorithm 2. From Figure 16, we see that similar to $q$-FFL, TERM ($t = 0.1$) can also significantly promote the accuracy on the worst device while maintaining the overall performance. The statistics of the accuracy distribution are reported in Table 7 below.

Table 7: Both $q$-FFL and TERM can encourage more uniform accuracy distributions across the devices in federated networks while maintaining similar average performance.

| objectives | test accuracy | | | |
|---|---|---|---|---|
| | average | worst 10% | best 10% | standard deviation |
| FedAvg | 0.853 (0.173) | 0.421 (0.016) | **0.951** (0.008) | 0.173 (0.003) |
| $q$-FFL ($q = 5$) | 0.862 (0.065) | **0.704** (0.033) | 0.929 (0.006) | **0.064** (0.011) |
| TERM ($t = 0.1$) | 0.853 (0.061) | **0.707** (0.021) | 0.926 (0.006) | **0.061** (0.006) |

**Improving generalization via variance reduction.** We compare TERM (applied at the class-level as in (4), with logistic loss) with robustly regularized risk (RobustRegRisk) as in (Namkoong & Duchi, 2017) on the HIV-1 (Dua & Graff, 2019; Rögnvaldsson, 2013) dataset originally investigated

by (Namkoong & Duchi, 2017). We examine the accuracy on the rare class ($Y = 0$), the common class ($Y = 1$), and overall accuracy.

The mean and standard error of accuracies are reported in Table 8. RobustRegRisk and TERM offer similar performance improvements compared with other baselines, such as linear SVM, Learn-Rewight (Ren et al., 2018), FocalLoss (Lin et al., 2017), and HRM (Leqi et al., 2019). For larger $t$, TERM achieves similar accuracy in both classes, while RobustRegRisk does not show similar trends by sweeping its hyperparameters. It is common to adjust the decision threshold to boost the accuracy on the rare class. We do this for ERM and RobustRegRisk and optimize the threshold so that ERM$_+$ and RobustRegRisk$_+$ result in the same validation accuracy on the rare class as TERM ($t = 50$). TERM achieves similar performance to RobustRegRisk$_+$, without the need for an extra tuned hyperparameter.

Table 8: TERM ($t = 0.1$) is competitive with strong baselines in generalization. TERM ($t = 50$) outperforms ERM$_+$ (with decision threshold changed for providing fairness) and is competitive with RobustRegRisk$_+$ with no need for extra hyperparameter tuning.

| objectives | accuracy ($Y = 0$) | | accuracy ($Y = 1$) | | overall accuracy (%) | |
|---|---|---|---|---|---|---|
| | train | test | train | test | train | test |
| ERM | 0.841 (.005) | 0.822 (.009) | 0.971 (.000) | 0.966 (.002) | 0.944 (.000) | 0.934 (.003) |
| Linear SVM | 0.873 (.003) | **0.838** (.013) | 0.965 (.000) | 0.964 (.002) | 0.951 (.001) | **0.937** (.004) |
| LearnReweight (Ren et al., 2018) | 0.860 (.004) | **0.841** (.014) | 0.960 (.002) | 0.961 (.004) | 0.940 (.001) | 0.934 (.004) |
| FocalLoss (Lin et al., 2017) | 0.871 (.003) | **0.834** (.013) | 0.970 (.000) | 0.966 (.003) | 0.949 (.001) | **0.937** (.004) |
| HRM (Leqi et al., 2019) | 0.875 (.003) | **0.839** (.012) | 0.972 (.000) | 0.965 (.003) | 0.952 (.001) | **0.937** (.003) |
| RobustRegRisk (Namkoong & Duchi, 2017) | 0.875 (.003) | **0.844** (.010) | 0.971 (.000) | 0.966 (.003) | 0.951 (.001) | **0.939** (.004) |
| TERM ($t = 0.1$) | 0.864 (.003) | **0.840** (.011) | 0.970 (.000) | 0.964 (.003) | 0.949 (.001) | **0.937** (.004) |
| ERM$_+$ (thresh = 0.26) | 0.943 (.001) | 0.916 (.008) | 0.919 (.001) | 0.917 (.003) | 0.924 (.001) | 0.917 (.002) |
| RobustRegRisk$_+$ (thresh=0.49) | 0.943 (.000) | 0.917 (.005) | 0.928 (.001) | **0.928** (.002) | 0.931 (.001) | **0.924** (.001) |
| TERM ($t = 50$) | 0.942 (.001) | 0.917 (.005) | 0.926 (.001) | **0.925** (.002) | 0.929 (.001) | **0.924** (.001) |

## J  EXPERIMENTAL DETAILS

We first describe the datasets and models used in each experiment presented in Section 5, and then provide a detailed setup including the choices of hyperparameters. All code and datasets are publicly available at github.com/litian96/TERM.

### J.1  DATASETS AND MODELS

We apply TERM to a diverse set of real-world applications, datasets, and models.

In Section 5.1, for regression tasks, we use the drug discovery data extracted from (Diakonikolas et al., 2019) which is originally curated from (Olier et al., 2018) and train linear regression models with different losses. There are 4,085 samples in total with each having 411 features. We randomly split the dataset into 80% training set, 10% validation set, and 10% testing set. For mitigating noise on classification tasks, we use the standard CIFAR-10 data and their standard train/val/test partitions along with a standard inception network (Szegedy et al., 2016). For experiments regarding mitigating noisy annotators, we again use the CIFAR-10 data and their standard partitions with a ResNet20 model. The noise generation procedure is described in Section 5.1.

In Section 5.2, for fair PCA experiments, we use the complete Default Credit data to learn low-dimensional approximations and the loss is computed on the full training set. We follow the exact data processing steps described in the work (Samadi et al., 2018) we compare with. There are 30,000 total data points with 21-dimensional features (after preprocessing). Among them, the high education group has 24,629 samples and the low education group has 5,371 samples. For class imbalance experiments, we directly take the unbalanced data extracted from MNIST (LeCun et al., 1998) used in (Ren et al., 2018). When demonstrating the variance reduction of TERM, we use the HIV-1 dataset (Rögnvaldsson, 2013) as in (Namkoong & Duchi, 2017) and randomly split it into 80% train, 10% validation, and 10% test set. There are 6,590 total samples and each has 160 features. We report results based on five such random partitions of the data. We train logistic regression models (without any regularization) for this binary classification task for TERM and the baseline methods. We also investigate the performance of a linear SVM.

In Section 5.3, the HIV-1 data are the same as that in Section 5.2. We also manually subsample the data to make it more imbalanced, or inject random noise, as described in Section 5.3.

### J.2  HYPERPARAMETERS

**Selecting $t$.** In Section 5.2 where we consider positive $t$'s, we select $t$ from a limited candidate set of $\{0.1, 0.5, 1, 5, 10, 50, 100, 200\}$ on the held-out validation set. For initial robust regression experiments, RMSE changed by only 0.08 on average across t; we thus used $t = -2$ for all experiments involving noisy training samples (Section 5.1 and Section 5.3).

**Other parameters.**  For all experiments, we tune all other hyperparameters (the learning rates, the regularization parameters, the decision threshold for ERM$_+$, $\rho$ for (Namkoong & Duchi, 2017), $\alpha$ and $\gamma$ for focal loss (Lin et al., 2017)) based on a validation set, and select the best one. For experiments regarding focal loss (Lin et al., 2017), we select the class balancing parameter ($\alpha$ in the original focal loss paper) from `range(0.05, 0.95, 0.05)` and select the main parameter $\gamma$ from $\{0.5, 1, 2, 3, 4, 5\}$. We tune $\rho$ in (Namkoong & Duchi, 2017) such that $\frac{\rho}{n}$ is selected from $\{0.5, 1, 2, 3, 4, 5, 10\}$ where $n$ is the training set size. All regularization parameters including regularization for linear SVM are selected from $\{0.0001, 0.01, 0.1, 1, 2\}$. For all experiments on the baseline methods, we use the default hyperparameters in the original paper (or the open-sourced code).

We summarize a complete list of main hyperparameter values as follows.

*Section 5.1:*

- Robust regression. The threshold parameter $\delta$ for Huber loss for all noisy levels is 1, the corruption parameter $k$ for CRR is: 500 (20% noise), 1000 (40% noise), and 3000 (80% noise); and TERM uses $t = -2$.
- Robust classification. The results are all based on the default hyperparameters provided by the open-sourced code of MentorNet (Jiang et al., 2018), if applicable. We tune the $q$ parameter for

generalized cross entropy (GCE) from $\{0.4, 0.8, 1.0\}$ and select a best one for each noise level. For TERM, we scale $t$ linearly as the number of iterations from 0 to -2 for all noise levels.

- Low-quality annotators. For all methods, we use the same set of hyperparameters. The initial step-size is set to 0.1 and decayed to 0.01 at epoch 50. The batch size is 100.

*Section 5.2:*

- Fair PCA. We use the default hyperparameters and directly run the public code of (Samadi et al., 2018) to get the results on the min-max fairness baseline. We use a learning rate of 0.001 for our gradient-based solver for all target dimensions.

- Handling class imbalance. We take the open-sourced code of LearnReweight (Ren et al., 2018) and use the default hyperparameters for the baselines of LearnReweight, HardMine, and ERM. We implement focal loss, and select $\alpha = 0.05, \gamma = 2$.

- Variance reduction. The regularization parameter for linear SVM is 1. $\gamma$ for focal loss is 2. We perform binary search on the decision thresholds for $\text{ERM}_+$ and $\text{RobustRegRisk}_+$, and choose 0.26 and 0.49, respectively.

*Section 5.3:*

- We tune the $q$ parameter for GCE based on validation data. We use $q = 0, 0, 0.7, 0.3$ respectively for the four scenarios we consider. For RobustlyRegRisk, we use $\frac{\rho}{n} = 10$ (where $n$ is the training sample size) and we find that the performance is not sensitive to the choice of $\rho$. For focal loss, we tune the hyperparameters for best performance and select $\gamma = 2$, $\alpha = 0.5, 0.1, 0.5$, and 0.2 for four scenarios. We use $t = -2$ for TERM in the presence of noise, and tune the positive $t$'s based on the validation data. In particular, the values of tilts under four cases are: (0, 0.1), (0, 50), (-2, 5), and (-2, 10) for $\text{TERM}_{sc}$ and (0.1, 0), (50, 0), (1, -2) and (50, -2) for $\text{TERM}_{ca}$.

# K    DISCUSSION

Our proposed work provides an alternative to empirical risk minimization (ERM), which is ubiquitous throughout machine learning. As such, our framework (TERM) could be widely used for applications both positive and negative. However, our hope is that the TERM framework will allow machine learning practitioners to easily modify the ERM objective to handle practical concerns such as enforcing fairness amongst subgroups, mitigating the effect of outliers, and ensuring robust performance on new, unseen data. One potential downside of the TERM objective is that if the underlying dataset is *not* well-understood, incorrectly tuning $t$ could have the unintended consequence of *magnifying* the impact of biased/corrupted data in comparison to traditional ERM. Indeed, critical to the success of such a framework is understanding the implications of the modified objective, both theoretically and empirically. The goal of this work is therefore to explore these implications so that it is clear when such a modified objective would be appropriate.

In terms of the use-cases explored with the TERM framework, we relied on benchmark datasets that have been commonly explored in prior work (e.g., Samadi et al., 2018; Tantipongpipat et al., 2019; Yang et al., 2010; Yu et al., 2012). However, we note that some of these common benchmarks, such as cal-housing (Pace & Barry, 1997) and Credit (Yeh & Lien, 2009), contain potentially sensitive information. While the goal of our experiments was to showcase that the TERM framework could be useful in learning fair representations that suppress membership bias and hence promote fairer performance, developing an understanding for—and removing—such membership biases requires a more comprehensive treatment of the problem that is outside the scope of this work.

