# OpenReview forum: "Tilted Empirical Risk Minimization"
_ICLR.cc/2021/Conference — ICLR 2021 Poster_

### Official Review · AnonReviewer4 · 2020-10-27
**Review of "Tilted Empirical Risk Minimization"**

**Rating:** 8
**Confidence:** 4

**Review:**



  *  Summary of the paper.
  The paper presents a modification of the empirical risk minimization (ERM) framework in order to obtain more robust or fairer results, using tilted objective. The paper exhibit several properties of the tilted empirical risk minimization (TERM) algorithm applied to classification and regression, TERM exhibits a trade-off between average loss and max loss for instance and this trade-off can be handled with a parameter t that control how robust one wants to be or how fair one wants to be. The claims are supported by a number of numerical illustrations that show the practical efficiency of the method for diverse tasks.

  * Strong points: the article is entirely reproducible as the author furnishes a well documented code. The figures and experiments made in the article provide a good illustration of the algorithm and a few theoretical lemma come to help understand the algorithm better.

  * Weak points: Use of 80% outliers in Table 1 without further explications. I think there is too much material, you made a lot of different experiments and it would have been great to have less experiments but more explications. There is no theoretical risk bound that would give us the efficiency of TERM, for instance in a corrupted setting or with respect to a fairness loss.

  *  Recommendation.
    I vote for accept. The algorithm seems efficient and easy to implement and it allows the user a broad choice of different extensions of the basic learning framework in particular to fairness and robustness in classification and regression.

**Most of my review will be about the robustness part of the article because it is a subject I am familiar with.**

  *  Questions:
    * It is very weird that TERM works in Table 1 even when there are 80% outliers , as my understanding of outliers is that they must be in minority. I would say that if 80% of the points are outliers, maybe the 20% points are in fact the outliers and I don't see a practical application where the inliers would be in minority. It seems counterintuitive. Can you explain that ? This is even more counterintuitive in classification where the error is smaller when the corruption is 80% than when it is 40%. I am not sure what is the task and what it is that your algorithm do in the 80% noise context. All the definitions of noise/outliers that I know of suppose that the proportion of outliers is smaller than 50%. In fact I think the problem with 80% outliers is theoretically impossible except when doing list decoding (see for instance the article "List-Decodable Robust Mean Estimation and Learning Mixtures of Spherical Gaussians" by Diakonikolas, Kane and Stewart).

    * Why did you not compare your algorithms with scikit-learn algorithms or scikit-learn-contrib algorithms ? I am thinking about HuberRegressor, RANSAC, TheilSenRegressor. To go further you could also test algorithms from scikit-lego for fairness or scikit-learn-extra for robust classification. To compare to a second party and not only your algorithms vs your algorithms.

    * Why did you only corrupt the labels in a robust classification task ? At least in the regression experiment, you could have corrupted the features and it works well. This is a very interesting property of your algorithm, there are not a lot of robust regression and classification algorithms that are robust to outliers in the feature space ! Is it a misunderstanding on my part or do you algorithm really work when feature space is corrupted ? I tested your algorithm on drug experiment with outliers in feature space and it worked. Huber and L1 did not work in this context.

    * Is the TERM problem convex when t<0 ? As it is, there is no reason that your algorithm will always converge to a global minimum.

  * Additional Feedback.
    * The 80% noise in Table 1 really bugged me when I read your article, maybe you may want to explain more or to remove it as it can cause misunderstandings I think.
    * When comparing robust methods, (Table 1), the authors did not compare their methods to the mainstream algorithms like RANSAC or Theil-Sen regression. It would have been interesting to do so.
    * The classification task considered by the authors in Table 1 (CIFAR-10) is not easy to interpret. What is an outlier for a neural network ? For a linear classifier, an outlier is readily defined for a very non-linear classifier; this is not so easy because most neural networks will be so nonlinear that the introduction of outliers in the training dataset will not change the performances of the algorithm.
    * It would have been interesting to compare TERM to other robust classification algorithms on an easier dataset in low dimension to exhibit the comparative performances similarly to what is done for regression.
    * You talk of "minimax" in a way that is unusual. Most of the time, at least in theoretical ML, minimax algorithms refer to algorithms which attain the optimal rate of convergence (see Section 14.1 in "A Probabilistic Theory of Pattern Recognition" by Devroye, Gyorfi and Lugosi). I think this is not what you mean, what you used I call it minmax. I don't know if yours is a common use of the term and if this is the case, sorry for this comment and don't take it into account.
    * Typo: Section 5.3 inn -> in.
    * The fact that the objective is strongly convex when t>0 is fairly important (this proves the convergence), I think that you should include the whole proof and not an abridged version of it (see proof Lemma 3).

---

> ### Author Response · Authors · 2020-11-17
> **Response to Reviewer #4**
>
> We appreciate the reviewer's detailed reviews for improving the draft.
>
> **[Convexity for t<0]** TERM becomes nonconvex for sufficiently negative t; we state this point in Appendix H (the ‘Convergence with t’ section).
>
> **[80% noise in robust regression and classification]**
>
> The outliers we consider in the experiments are random noisy samples without any structure. In such cases, if 20% of the clean data have a clear structure and the remaining are random noise, it is possible that the model can still discover the underlying structure of the clean data. We were motivated to look at this application and the extreme noise regimes (> 50% noise ratio) based on prior work in classification, e.g., https://arxiv.org/pdf/1712.05055.pdf, https://arxiv.org/pdf/1805.07836.pdf, which we compared with in our submission.
>
> We do agree with the reviewer that it is likely not possible to mitigate outliers if the 80% outliers were structured or adversarial. We are not even sure whether one could recover noise levels smaller than 50% in the presence of adversarial noise. We have added a discussion regarding this point in the main text in Section 5 (‘robust regression’).
>
> To further highlight the differing performance under structured vs. unstructured noise, we have also added some toy examples to Appendix I.2  (Figure 11, 12, 13). For unstructured random noise, we generate synthetic data, and run regression and classification tasks on them under noise ratio 0%,  20%, 40%, and 80%. These results show that TERM with t<0 is fairly robust under all noise levels (while the performance does drop at 80% noise, it can still learn useful information, and achieves much lower error than ERM). In contrast (and to the reviewer’s point), we have also constructed cases based on structured noise in Figure 14 where TERM in fact fits to the noisy samples and ignores the ‘clean’ samples for large noise values (80%), as would be expected.
>
> Regarding your concern on the interpretability of the noise in CIFAR10, we have conducted experiments on synthetic data for robust classification for a binary case using logistic regression as the loss function (Figure 13, Appendix I.2). Similarly, we see that TERM (t=-2) is fairly robust. We have added discussions on this point in Section 5. We also compare with the generalized cross entropy baseline (GCE), which we also compared against on the CIFAR10 dataset. We note that in this toy example, GCE is as good as TERM with t=-2. On the real dataset CIFAR 10, TERM outperforms GCE (Table 2).
>
> **[Comparison with Scikit-learn]** We implemented the baselines ourselves (as opposed to using a library like scikit-learn) in order to remove implementation differences/randomness induced by the complicated abstractions provided by a third-party library, and to keep everything else (e.g., the optimizer, the data shuffling, etc) fixed except the objective for a fair comparison. Additionally, we compared with a number of recent, state-of-the-art methods, which we would expect to be superior to the default methods in scikit-learn. However, as per your suggestion, we have conducted experiments comparing to robust methods provided by sklearn on robust regression. The results are shown in the table below.
>
>
> |scikit-learn methods |20% noise | 40% noise |80% noise|
> |-----|-----|-----|-----|
> |LinearRegressor (L2) |2.96 (.18) |2.89 (.35) |4.09 (.61)|
> |HuberRegressor | 1.40 (.68) | 2.14 (.14) |3.72 (.55) |
> |RANSAC | 2.63 (.14)  | 2.87 (.33)  | 4.07 (.62)|
>
> As we can see, the RANSAC method is worse than Huber loss in this case, and all methods perform considerably worse than TERM. Moreover, the Theil-Sen regressor in scikit-learn fails to finish running within a fairly long time (1000x of the L2 regressor). We suspect this is due to the fact that this algorithm does not scale well with the number of samples and the feature dimension.
>
> **[Feature corruption experiments]** We greatly appreciate the reviewer’s careful evaluation, which included running our code in a different setting. We agree that TERM can also obtain good performance with feature noise, and we do have such experiments in the appendix (Table 5). We are happy to bring these or related feature corruption experiments from the appendix into the main text if the reviewer thinks that would add value to the paper.
>
> **[Formal convergence results]** We added a theorem (Theorem 13) on the convergence rates for batch TERM. Please see our response to all reviewers for details.
>
> **[Others]** We believe minimax is commonly used to refer to any objective minimizing the worst loss, as applicable to a multitude of problems (see e.g. https://en.wikipedia.org/wiki/Minimax); however, we are happy to switch to min-max to avoid any confusion.

---

### Official Review · AnonReviewer2 · 2020-10-28
**Interesting analysis with a broad range of applications**

**Rating:** 6
**Confidence:** 3

**Review:**

This work analyzes the LogSumExp aggregated loss (named tiled empirical risk minimization, or TERM, in the paper). It provides several general properties of the loss, such as its relation to min/avg/max-loss, and interpretations of different trade-offs. Empirically, it is shown that TERM can be applied to a diverse set of problems, including robust optimization, fairness and generalization.

Strength:
1. Provide theoretical analysis on the properties of TERM
2. Various experiments to showcase the usefulness of TERM as objective

Weakness:
1. Unclear convergence of the optimization procedure
2. Missing literature

Details:
1. The LogSumExp has been extensively studied in geometric programming (Calafiore and El Ghaoui, 2014, Sec.9.7) and boosting (Mason et al., 2000; Shen and Li, 2010). The current manuscript does have some novel interpretations such as the trade-off between avg-min losses. However, I am not an expert in this field, thus not sure how much the new analysis will contribute to the community (or the analysis may exist somewhere). Additionally, LogSumExp is called "tilted" without explanation. Why not call it LogSumExp, which is well-known?

Ref:
- Calafiore, G.C. and El Ghaoui, L., 2014. Optimization models. Cambridge university press.
- Mason, L., Baxter, J., Bartlett, P.L. and Frean, M.R., 2000. Boosting algorithms as gradient descent. In Advances in neural information processing systems (pp. 512-518).
- Shen, C. and Li, H., 2010. On the dual formulation of boosting algorithms. IEEE Transactions on Pattern Analysis and Machine Intelligence, 32(12), pp.2216-2231.

2. Some of the technical details are missing.
- What is the range of t in Fig.1?
- The interpretation 2 is constrained to be t<0 for avg-min trade-off (and t>0 for avg-max, as shown empirically in Fig.9). This should be clear in the informal statement to avoid confusion. Also, the interpretation 3 is valid for t>0.
- Algorithm 2 claims that t is temperature, but in fact, t is more like the inverse of temperature in common sense (consider softmax).

3. Convergence of the optimization procedure is not convincing. Since Algo.2 is using a non-trivial averaging for the normalization term, the convergence of the stochastic version is unclear. It seems that the convergence discussion in Appendix H is only for the batch version.

---

> ### Author Response · Authors · 2020-11-17
> **Response to Reviewer #2**
>
> We thank the reviewer for the helpful comments.
>
> **[Convergence of TERM]** We have added Theorem 13 on the convergence guarantees for batch TERM, and we empirically demonstrate the convergence of stochastic TERM. Please see our response to all reviewers for more details along these lines.
>
> **[LogSumExp v.s. Tilting]** ‘LogSumExp’ objectives consider exponential smoothing to approximate the max. While we cited related literature on smoothly approximating the maximum in our original submission, we now also explicitly mention the term ‘LogSumExp’ to make the connection clear, and have added the references you mentioned. The term “tilted” comes from the idea of “tilting” in the importance sampling literature. In contrast to LogSumExp, which conceptually focuses on positive t’s, exponential tilting considers both positive and negative t’s. We therefore view “tilting” as a more general notion, and believe it more aptly describes the objective we investigate (which encompasses both positive/negative t).
>
> The existing analysis on LogSumExp is certainly relevant to TERM and we make these connections explicit in the appendix where we present our results (e.g., the strong convexity property is known). However, we note that the analyses on the properties of the TERM objective and its solutions are new (e.g., the tradeoffs between max-loss, and avg-loss or the variance reduction property), and the connections made with superquantile optimization are also entirely novel.
>
> **[Other details]**
> (1) The t's in Figure 1 are in the range [-10,10], which roughly matches the performance of t $\in (-\infty, +\infty)$ for these toy problems.
> (2) In the text, we explicitly state that the avg/max loss tradeoff corresponds to positive t’s and the avg/min loss tradeoff corresponds to negative t’s. In the informal statement presented under the text, the non-increasing of max loss holds for all t’s, and the non-decreasing of avg loss applies to only positive t’s. We have revised this statement. Thanks for pointing this out.
> (3) We have changed the temperature t to temperature 1/t to match convention, as per your suggestion.

---

> > ### Comment · AnonReviewer2 · 2020-11-24
> > **Additional comments**
> >
> > Thank you for the explanations.
> >
> > The convergence of the stochastic version is crucial since it is more practical for large scale datasets. Traditional ERM-based algorithms can handle this, so it would be nice for TERM to have similar converging algorithms.
> >
> > One more comment on the "80% outlier" issue from R4. I agree that the term "outliers" can be misleading as they are usually in minority and erroneous in certain ways. It might be less confusing if they are called "noisy samples" instead.

---

> > > ### Author Response · Authors · 2020-11-25
> > > **Response to additional comments**
> > >
> > > Thank you for your additional feedback. We hope our response has addressed the rest of the reviewer's comments in the last round, and we respond to additional comments below.
> > >
> > > **[convergence guarantees for stochastic TERM]** In this work, our aim was to rigorously understand the properties of the TERM objective and its solutions, and demonstrate TERM's generality for a wide range of ML applications. Based on the competitive empirical performance of TERM and the stochastic method we used in a subset of the experiments, as well as the direct comparisons between the stochastic and batch method (Figure 7), we agree that more formal convergence guarantees for the stochastic case would be an interesting direction of future work. In comparison to traditional ERM, these guarantees are slightly more complex due to the fact that the stochastic gradient estimator is biased, thus requiring additional machinery beyond what is typically needed for stochastic solvers for vanilla ERM.
> > >
> > > **[noisy outliers vs. noisy samples]** Thank you for this suggestion; we will use the word 'noisy samples' rather than 'outliers' to avoid confusion.

---

### Official Review · AnonReviewer3 · 2020-10-29
**This paper provides a unified framework for solving a bunch of issues in ERM.**

**Rating:** 6
**Confidence:** 3

**Review:**

This paper considers a unified framework named TERM for addressing a bunch of problems arising in the simple averaged empirical minimization. By leveraging the key hyper-parameter t in the TERM loss, it can recover the original average loss and approximate robust loss, min/max loss, and the superquantile loss, etc. The authors also propose gradient-based optimization algorithms for solving the TERM problem.

One thing that I do not understand very well is the paragraph under Lemma 1. Why is it necessary that outliers can cause a large (positive t) or small (negative t)  losses? Note that outliers can be arbitrary, say adversarial.

Also, do you have numerical issues for large enough t?

Is it possible to show certain convergence results of the algorithms for solving the TERM? Especially, the TERM has the nice property that it is always smooth (depending on the value of t).

Overall, the TERM seems to be a good unification of different losses used in machine learning society for different purposes. The theoretical justifications also look reasonable and informative. In addition, the authors conduct a series of experiments to show the good performance of TERM for different tasks such as robustness to outliers, handling imbalance, and improving generalization, etc.

---

> ### Author Response · Authors · 2020-11-17
> **Response to Reviewer #3**
>
> We thank the reviewer for the valuable feedbacks.
>
> **[Outlier definition]** The outliers we consider are noisy samples with large losses, not adversarial examples; this notion of ‘outlier’ is standard in other literature, e.g., this interpretation motivates the use of the median for robust machine learning. To avoid confusion, we have added discussions around random unstructured noise v.s. adversarial noise in Section 5. Please also see our response to Reviewer #4.
>
> **[Numerical issues for large t’s]** We do not encounter numerical issues when using the batch solver. However, we agree that we could face such issues when using the stochastic variant. We developed the stochastic solver using a tilted average to estimate the gradients to mitigate this potential issue, and we empirically show that the stochastic method achieves good performance in practice for a number of applications (specifically, for our experiments on robust classification, low-quality annotators, class imbalance, and fair federated learning).
>
> **[Convergence of TERM]** We have added Theorem 13 on the convergence guarantees for batch TERM, and we empirically demonstrate the convergence of stochastic TERM. Please see our response to all reviewers for more details along these lines.

---

### Official Review · AnonReviewer1 · 2020-10-29
**The paper explore relevant applications of exponential smoothing in several ML problems. It is well written and should be interesting for ML practioners.**

**Rating:** 6
**Confidence:** 3

**Review:**

The paper provide nice discussions of a simple modification of ERM paradigm. Mainly, it consists in applying an exponential smoothing to the loss function. The authors provide several interpretations and connexions with (robustness/fairness/quantile regression etc...) literature and show how the TERM can be adapted to such problem. The paper is well written, contains pedagogical illustrations and detailed properties of TERM.

- The success of Term heavily rely on a *magical* tuning of the parameter $t$ depending on the application. Grid search was used in this paper which considerably increases the complexity of the algorithm without necessarily improving significantly the accuracy when compared to competitors.

- The experiments does not report standard deviation in the train/test splitting. Averaging the test accuracy after several random splitting would be beneficial for clarity.

- The classical ERM formulation is written as sum of functions, which allows several advanced stochastic optimization algorithm. Such structure is destroyed in the tilted formulation.

- In Assumption 2, isn't it too restrictive to assume that $f(x_i, \cdot)$ does not have critical point (which does not seems true for quadratic loss)?

- To help the reader, the authors might recall on which quotient the L'hopital rule is applied (after verification of assumptions)

- In equation (87), the limit and derivation are permuted without justification (uniform convergence might be needed). Same for Eq (90)

- Should be interesting to compare with recent robust estimation such as median of means or robust gradient descent (Holland and Ikeda, 2019).

- Should be also interesting to know how the generalization bound (and sample complexity) of TERM compare with the one of ERM wrt $t$.

---

> ### Author Response · Authors · 2020-11-17
> **Response to Reviewer #1**
>
> We thank the reviewer for the suggestions to improve the paper.
>
> **[Tuning t]** The goal of our analyses is precisely to highlight the effect of 't' on TERM, so that this tuning does not need to be magical; our paper provides numerous theoretical and empirical insights that explain the effect of positive/negative values of 't'. In terms of our experiments, as discussed in Section 5, we select t by performing simple grid search on a handful of values of t for positive t examples, and use t=-2 across all experiments involving negative t. Importantly, we note that all the methods we compare against (other than vanilla ERM, which performs considerably worse) have at least one hyperparameter that requires tuning. TERM (with just a single hyperparameter) scales similarly as ERM. Therefore, it is in fact more lightweight than most of the competitors we compare against (we discuss this throughout Section 5).
>
> **[Train/test splits for experiments]** We report standard error for the experiments in Table 1 and Table 3 based on multiple runs with different train/test splits, as these datasets don’t come with a predefined train/test split. However, for CIFAR10 (Table 2 and Figure 3), which has a standard and widely-used train/test split, we use the standard train/test split for evaluation. Similarly, for the results in Figure 5, we use the exact train/test partition used by the baseline methods to directly compare to this prior work.
>
> **[Stochastic solvers]** While TERM does not maintain additivity of individual losses, the gradient is a weighted sum of the ERM gradient (see Section 2, Lemma 1), which still allows us to use stochastic methods to solve it (see Algorithms 2 & 4). Further, for a specific t, optimizing TERM is equivalent to optimizing $\frac{1}{N} \sum_i e^{t f(x_i; \theta)}$ (removing the log part in the objective), which preserves additivity.
>
> **[Comparison with other works]** Thanks for this suggestion; we have added a citation to the robust gradient descent method (Holland and Ikeda, 2019) in our related work. We have also performed additional experiments to test robust gradient descent on our robust regression application. We take their open-source code on Github and use the suggested hyperparameters. The results are shown in the following table:
>
> | method    | 20% noise     | 40% noise  | 80% noise |
> |------------------|--------------|-------------|--------------|
> | Robust GD    | 1.62 (.04) | 2.39 (.04) | 4.18 (.07)|
>
> Robust GD does not achieve higher test accuracy than the baselines we compare with.
>
> **[Generalization of TERM]** We agree with the reviewer that generalization with respect to t would be an interesting direction of future work; we will mention this in the conclusion.
>
> **[Other technical details]** We clarify the reviewer’s concerns regarding a few technical details here.
>
> (1) The assumption that there doesn’t exist a $\theta$ such that $\nabla f(x_i; \theta )$ are all zero is not restrictive in the convex setting, which is the focus of the theoretical study in this paper. For example, for the loss in our first toy example in Figure 1, the assumption excludes the case where all data points $x_i$ are the same.
>
> (2) L’hopital’s rule is applied at $t$ and $\log \left(\frac{1}{N} \sum_{i \in [N]} e^{t f(x_i; \theta)} \right)$, respectively. We have added a sentence to clarify this.
>
> (3) We can exchange between the limit and the derivative because the gradient of the objective with respect to $\breve{\theta}(\tau)$ is a weighted sum of the gradients with weights bounded by [0, 1]. Please see our revisions for a detailed derivation. Thanks for these suggestions to improve the clarity of our work.

---

### Author Response · Authors · 2020-11-17
**Response to all reviewers**

We thank all reviewers for their valuable feedback. We first address shared concerns, and then respond to specific comments from each reviewer. We have updated the paper (with revisions highlighted in blue) and we refer to this updated version in our responses.

**[Contributions]** In this work, we explore tilted empirical risk minimization (TERM) as a simple, unified framework to address various challenges with ERM. We analyze the properties and solutions of the TERM objective, and provide novel connections between TERM and superquantile methods. We develop efficient solvers for TERM and show via multiple real-world applications that TERM achieves competitive performance with state-of-the-art, problem-specific approaches. Our work aims to highlight the effectiveness and versatility of tilted objectives in machine learning.

**[Convergence of TERM]** Our batch solver for TERM achieves the same convergence rates (differing by a constant linear with t) as standard gradient descent under the same assumptions. Our original submission included discussions on the convexity and smoothness of TERM in Appendix H. For completeness, we have also added Theorem 13 in Appendix H.1, which directly provides the convergence rates for the batch method. In terms of the stochastic solver, we have empirically demonstrated convergence by effectively applying it to various ML applications (specifically, our experiments on robust classification, low-quality annotators, class imbalance, and fair federated learning). We have also validated the convergence results on toy problems, where it is computationally feasible to run the batch method for comparison (Figure 7). Given the promising empirical performance, we agree with the reviewers that a theoretical analysis of the stochastic solver would be an interesting direction of future work; we highlight this in the conclusion.

---

### Author Response · Authors · 2021-03-10
**Code link**

https://github.com/litian96/TERM

---

### Comment · ~Seong_Hyeon_Park2 · 2021-04-22
**Typos at Eq. (19), (21).**

Thank you for contributing this great work to society. While reading through the paper, I found that the inequalities at Eq. (19) and Eq. (21) might need to be inverted. Please check. Thank you.

---

> ### Author Response · Authors · 2021-04-23
> **Reply**
>
> Thanks for catching the typos! We have reversed the inequality directions in Eq (19) and Eq (21) in the updated version, which do not affect anything else in the proof.

---

### Decision · Program_Chairs · 2021-01-07
**Final Decision**

**Decision:**

Accept (Poster)

**Comment:**

Dear Authors,

Thank you very much for your detailed feedback to the initial reviews and also for further answering additional questions raised by a reviewer. Your effort has been certainly contributed to clarifying some of the concerns raised by the reviewers and improving their understanding of this paper.

Overall, all the reviewers found a merit in this paper and thus I suggest its acceptance. However, as Reviewer #2 suggested, investigating the convergence in the stochastic case is very important. More discussion on this would be a valuable addition to the paper, which the authors can incorporate in the final version.